# Learning Confidence Sets using Support Vector Machines

**Wenbo Wang**
Department of Mathematical Sciences
Binghamton University
Binghamton, NY 13902
wang2@math.binghamton.edu

**Xingye Qiao***
Department of Mathematical Sciences
Binghamton University
Binghamton, NY 13902
qiao@math.binghamton.edu

## Abstract

The goal of confidence-set learning in the binary classification setting [14] is to construct two sets, each with a specific probability guarantee to cover a class. An observation outside the overlap of the two sets is deemed to be from one of the two classes, while the overlap is an ambiguity region which could belong to either class. Instead of plug-in approaches, we propose a support vector classifier to construct confidence sets in a flexible manner. Theoretically, we show that the proposed learner can control the non-coverage rates and minimize the ambiguity with high probability. Efficient algorithms are developed and numerical studies illustrate the effectiveness of the proposed method.

## 1 Introduction

In binary classification problems, the training data consist of independent and identically distributed pairs $(\boldsymbol{X}_i, Y_i)$, $i = 1, 2, ..., n$ drawn from an unknown joint distribution $P$, with $\boldsymbol{X}_i \in \mathcal{X} \subset \mathbb{R}^p$, and $Y_i \in \{-1, 1\}$. While the misclassification rate is a good assessment of the overall classification performance, it does not directly provide confidence for the classification decision. Lei [14] proposed a new framework for classifiers, named classification with confidence, using notions of *confidence* and *efficiency*. In particular, a classifier $\phi(\boldsymbol{x})$ therein is set-valued, i.e., the decision may be $\{-1\}$, $\{1\}$, or $\{-1, 1\}$. Such a classifier corresponds to two overlapped regions in the sample space $\mathcal{X}$, $C_{-1}$ and $C_1$, and they satisfy that $C_{-1} \cup C_1 = \mathcal{X}$. With these regions, we have the set-valued classifier

$$\phi(\boldsymbol{x}) = \begin{cases} \{-1\}, \text{when } \boldsymbol{x} \in C_{-1} \backslash C_1 \\ \{1\}, \text{when } \boldsymbol{x} \in C_1 \backslash C_{-1} \\ \{-1, 1\}, \text{when } \boldsymbol{x} \in C_{-1} \cap C_1 \end{cases}.$$

Those points in the first two sets are classified to a single class as by traditional classifiers. However, those in the overlap receive a decision of $\{-1, 1\}$, hence may belong to either class. When the option of $\{-1, 1\}$ is forbidden, the set-valued classifier degenerates to a traditional classifier.

Lei [14] defined the notion of *confidence* as the probability $100(1-\alpha_j)\%$ that set $C_j$ covers population class $j$ for $j = \pm 1$ (recalling the confidence interval in statistics). The notion of *efficiency* is opposite to *ambiguity*, which refers to the size (or probability measure) of the overlapped region named the *ambiguity region*. In this framework, one would like to encourage classifiers to minimize the ambiguity when controlling the non-coverage rates. Lei [14] showed that the best such classifier, the Bayes optimal rule, depends on the conditional class probability function $\eta(\boldsymbol{x}) = P(Y = 1 | \boldsymbol{X} = \boldsymbol{x})$. Lei [14] then proposed to use the plug-in method, namely to first estimate $\eta(\boldsymbol{x})$ using, for instance, logistic regression, then plug the estimation into the Bayes solution. Needless to say, its empirical performance highly depends on the estimation accuracy of $\eta(\boldsymbol{x})$. However, it is well known that the

latter can be more difficult than mere classification [24, 9, 26], especially when the dimension $p$ is large [27].

Support vector machine [SVM; 5] is a popular classification method with excellent performance for many real applications. Fernández-Delgado et al. [7] compared 179 classifiers on 121 real data sets and concluded that SVM was among the best and most powerful classifiers. To avoid estimating the conditional class probability $\eta(\boldsymbol{x})$, we propose a support vector classifier to construct confidence sets by empirical risk minimization. Our method is more flexible as it takes advantage of the powerful prediction power of support vector machine.

We show in theory that the population minimizer of our optimization is to some extent equivalent to the Bayes optimal rule in [14]. Moreover, in the finite-sample case, our classifier can control both non-coverage rates while minimizing the ambiguity.

A closely related problem is the Neyman-Pearson (NP) classification [4, 19] whose goal is to find a boundary for a specific null hypothesis class. It aims to minimize the probability that an observation from the alternative class falls into this region (the type II error) while controlling the type I error, i.e., the non-coverage rate for the null class. See Tong et al. [22] for a survey. Our problem can be understood as a two-sided NP classification problem. Other related areas of work are conformal learning, set-valued classification, or classification with reject and refine options. See [21], [6], [22], [23], [11], [2] and [28].

The rest of the article is organized as follows. Some background information is provided in Section 2. Our main method is introduced in Section 3. A comprehensive theoretical study is conducted in Section 4, including the Fisher consistency and novel statistical learning theory. In Section 5, we present efficient algorithms to implement our method. The usefulness of our method is demonstrated using simulation and real data in Section 6. Detailed proofs are in the Supplementary Material.

## 2  Background and notations

We first formally define the problem and give some useful notations.

It is desirable to keep the ambiguity as small as possible. On the other hand, we would like as many class $j$ observations as possible to be covered by $C_j$. Consider predetermined non-coverage rates $\alpha_{-1}$ and $\alpha_1$ for the two classes. Let $P_{-1}$ and $P_1$ be the probability measure of $\boldsymbol{X}$ conditional on $Y = -1$ and $+1$. Conceptually, we formulate classification with confidence as the optimization below.

$$\min_{C_{-1}, C_1} P\left(C_{-1} \cap C_1\right) \quad \text{subject to } P_j(C_j) \geq 1 - \alpha_j, \ j = \pm 1, \quad C_{-1} \cup C_1 = \mathcal{X}. \tag{1}$$

Here the constraint that $P_j(C_j) \geq 1 - \alpha_j$ means that $100(1 - \alpha_j)\%$ of the observations from class $j$ should be covered by region $C_j$.

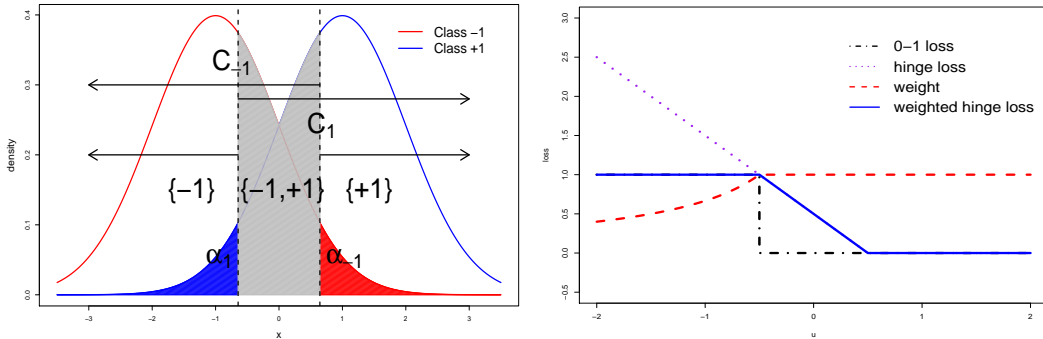

Figure 1: The left panel shows the two definite regions and the ambiguity region in the case of symmetric Gaussian distributions. The right penal illustrates the weight function (see Section 3).

Under certain conditions, the Bayes solution of this problem is: $C_{-1}^* = \{\boldsymbol{x} : \eta(\boldsymbol{x}) \leq t_{-1}\}$ and $C_1^* = \{\boldsymbol{x} : \eta(\boldsymbol{x}) \geq t_1\}$ with $t_{-1}$ and $t_1$ satisfying that $P_{-1}(\eta(\boldsymbol{X}) \leq t_{-1}) = 1 - \alpha_{-1}$ and $P_1(\eta(\boldsymbol{X}) \geq t_1) = 1 - \alpha_1$. A simple illustrative toy example with two Gaussian distributions on $\mathbb{R}$ is shown in Figure 1. The two boundaries are shown as the vertical lines, which lead to three decision

regions, $\{-1\}$, $\{+1\}$, and $\{-1, +1\}$. The non-coverage rate $\alpha_{-1}$ for class $-1$ is shown on the right tail of the red curve (similarly, $\alpha_1$ for class 1 on the left tail of blue curve.) In reality, the underlying distribution will be more complicated than a simple multivariate Gaussian distribution and the true boundary may be beyond linearity. In these cases, flexible approaches such as SVM will work better.

Confidence sets may be seen as equivalent to classification with reject options [11, 2, 10] via different parameterizations. The Bayes rule in this article is different from the Bayes rule in the literature of classification with reject options. In that context, the Bayes rule depends on a comparison between $\eta(\cdot)$ and a predetermined cost of rejection $d$. But it does not lead to a guarantee of the coverage probabilities for the corresponding confidence sets. Here instead, the cutoff for the Bayes rule is calibrated to achieve the desired coverage probabilities.

## 3 Learning confidence sets using SVM

To avoid estimating $\eta$, we propose to solve the empirical counterpart of (1) directly using SVM. Here, we present two variants of our method. We start with an original version to illustrate the basic idea. Then we introduce an improvement.

Unlike the regular SVM, the proposed classifier has two (not one) separating boundaries. They are defined as $\{\boldsymbol{x} : f(\boldsymbol{x}) = -\varepsilon\}$ and $\{\boldsymbol{x} : f(\boldsymbol{x}) = +\varepsilon\}$ where $f$ is the discriminant function, and $\varepsilon \geq 0$. The positive region $C_1$ is $\{\boldsymbol{x} : f(\boldsymbol{x}) \geq -\varepsilon\}$ and the negative region $C_{-1}$ is $\{\boldsymbol{x} : f(\boldsymbol{x}) \leq \varepsilon\}$. Hence when $-\varepsilon \leq f(\boldsymbol{x}) \leq \varepsilon$, observation $\boldsymbol{x}$ falls into the ambiguity region $\{-1, 1\}$.

Define $R(f, \varepsilon) = P(|Yf(\boldsymbol{X})| \leq \varepsilon)$ the probability measure of the ambiguity. We may rewrite problem (1) in terms of the function $f$ and threshold $\varepsilon$,

$$\min_{\varepsilon \in \mathbb{R}^+, f} R(f, \varepsilon), \quad \text{subject to} \quad P_j(Yf(\boldsymbol{X}) < -\varepsilon) \leq \alpha_j, \; j = \pm 1. \quad (2)$$

Replacing the probability measures above by the empirical measures, we can obtain,

$$\min_{\varepsilon \in \mathbb{R}^+, f} \frac{1}{n} \sum_{i=1}^{n} \mathbb{1}\{-\varepsilon \leq f(\boldsymbol{x_i}) \leq \varepsilon\}, \quad \text{subject to} \quad \frac{1}{n_j} \sum_{i:y_i=j} \mathbb{1}\{y_i f(\boldsymbol{x_i}) < -\varepsilon\} \leq \alpha_j, \; j = \pm 1.$$

It is easy to show that as long as the equalities in the constraints are achieved at the optimum, we can obtain the same minimizer if the objective function is changed to $\frac{1}{n} \sum_{i=1}^{n} \mathbb{1}\{y_i f(\boldsymbol{x_i}) - \varepsilon \leq 0\}$.

For efficient and realistic optimization, we replace the indicator function $\mathbb{1}\{u \leq 0\}$ in the objective function and constraints by the Hinge loss function $(1 - u)_+$. The practice of using a surrogate loss to bound the non-coverage rates has been widely used in the literature of NP classification, see [19]. To simplify the presentation, we denote $H_a(u) = (1 + a - u)_+$ as the $a$-Hinge Loss and it can be seen that $H_a(x)$ coincides with the original Hinge loss when $a = 0$. Our initial classifier can be represented by the following optimization:

$$\min_{\varepsilon \in \mathbb{R}^+, f} \frac{1}{n} \sum_{i=1}^{n} H_\varepsilon(y_i f(\boldsymbol{x_i})) + \lambda J(f), \quad \text{subject to} \quad \frac{1}{n_j} \sum_{i:y_i=j} H_{-\varepsilon}(y_i f(\boldsymbol{x_i})) \leq \alpha_j, \; j = \pm 1. \quad (3)$$

Here $J$ is a regularization term to control the complexity of the discriminant function $f$. When $f$ takes the linear form of $f(\boldsymbol{x}) = \boldsymbol{x}^T \boldsymbol{\beta} + b$, $J(f)$ can be $L_2$-norm $\|\boldsymbol{\beta}\|^2$ or $L_1$-norm $|\boldsymbol{\beta}|$.

In SVM, $yf(\boldsymbol{x})$ is called the functional margin, which measures the signed distance from $\boldsymbol{x}$ to the boundary $\{\boldsymbol{x} : f(\boldsymbol{x}) = 0\}$. Positive and large value of $yf(\boldsymbol{x})$ means the observation is correctly classified, and is far away from the boundary. In our situation, we compare $yf(\boldsymbol{x})$ with $+\varepsilon$ and $-\varepsilon$ respectively. If $yf(\boldsymbol{x}) < -\varepsilon$, then $\boldsymbol{x}$ is not covered by $C_y$ (hence is misclassified, in the classification language). On the other hand, if $yf(\boldsymbol{x}) \leq \varepsilon$, then $\boldsymbol{x}$ either satisfies that $yf(\boldsymbol{x}) < -\varepsilon$ as above, or falls into the ambiguity, which is why we try to minimize the sum of $H_\varepsilon(y_i f(\boldsymbol{x_i}))$.

By constraining $\sum_{y_i=j} H_{-\varepsilon}(y_i f(\boldsymbol{x_i}))$ for both classes, we aim to control the non-coverage rates. Since $H_{-\varepsilon}(u) \geq \mathbb{1}\{u < -\varepsilon\}$ (the latter indicates the occurrence of non-coverage) for negatively large $u$. It may be more conservative by using the Hinge loss than the indicator function $\mathbb{1}\{y_i f(\boldsymbol{x_i}) < -\varepsilon\}$ in the constraint to control the non-coverage rates. We alleviate this problem by imposing a weight $w_i$ to each observation in the constraint. In particular, this weight is chosen to be $w_i = \max\{1, H_{-\varepsilon}(y\hat{f}(\boldsymbol{x}))\}^{-1}$, where $\hat{f}$ is a reasonable guess of the final minimizer $f$. Our goal

is to weight the Hinge loss in the constraint, $w_i H_{-\varepsilon}(y_i f(\boldsymbol{x_i}))$, so that it approximates the indicator function $\mathbb{1}\{y_i f(\boldsymbol{x_i}) < -\varepsilon\}$. This may be illustrated by Figure 1 in which the blue bold line is the result of multiplying the weight (red dashed) by the Hinge loss (purple dotted), which is close to the indicator function (black dot-dashed). Note that by weighting the Hinge loss, the impact of those observations with very negatively large $u = yf(\boldsymbol{x})$ value is reduced to 1. The adaptive weighted version of our method changes constraint (3) to $\frac{1}{n_j} \sum_{i:y_i=j} w_i H_{-\varepsilon}(y_i f(\boldsymbol{x_i})) \leq \alpha_j, j = \pm 1$.

In practice, we adopt an iterative approach, and use the estimated $f$ from the previous iteration to calculate the weight for each observation at the current iteration. We start with equal weights for each observation, solve the optimization problem with the weights obtained in the last iteration, and then calculate the new weights for the next iteration. [25] first used this idea in their work of adaptively weighted large margin classifiers for the purpose of robust classification.

## 4 Theoretical Properties

In this section we study the theoretical properties of the proposed method. We start with population level properties in Section 4.1. In Section 4.2, we discuss the finite-sample properties using novel statistical learning theory.

### 4.1 Fisher consistency and excess risk

Assume that $P_{-1}$ and $P_1$ are continuous with density function $p_{-1}$ and $p_1$, and $\pi_j = P(Y = j)$ is positive for $j = \pm 1$. Moreover, $\eta(\boldsymbol{X})$ is continuous and has positive density function almost everywhere, and $t_{-1}$ and $t_1$ are quantiles of $\eta(\boldsymbol{X})$. They satisfies $P_{-1}(\eta(\boldsymbol{X}) \leq t_{-1}) = 1 - \alpha_{-1}$ and $P_1(\eta(\boldsymbol{X}) \geq t_1) = 1 - \alpha_1$. We need to make assumptions on the difficulty level of the classification task. In particular, the classification should be difficult enough so that overlapping regions is meaningful (otherwise, there will be almost no ambiguity even at small non-coverage rates.)

**Assumption 1.** $t_{-1} \geq \frac{1}{2} \geq t_1$.

**Assumption 2.** $\exists c > 0, t_{-1} - c \geq \frac{1}{2} \geq t_1 + c$.

Each assumption implies that the union of $C^*_{-1} = \{\boldsymbol{x} : \eta(\boldsymbol{x}) \leq t_{-1}\}$ and $C^*_1 = \{\boldsymbol{x} : \eta(\boldsymbol{x}) \geq t_1\}$ is $\mathcal{X}$. Otherwise, there will be a gap around the boundary $\{\boldsymbol{x} : \eta(\boldsymbol{x}) = 1/2\}$. It is easy to see that Assumption 2 is stronger than Assumption 1.

Fisher consistency concerns the Bayes optimal rule, which is the minimizer of problem (2). In (4) below, we replace the loss function in the objective function of (2) with risk under the Hinge loss.

$$\min R_H(f, \varepsilon), \quad \text{subject to } P_j(Yf(\boldsymbol{X}) < -\varepsilon) \leq \alpha_j, \ j = \pm 1, \tag{4}$$

where $R_H(f, \varepsilon) = E[H_\varepsilon(Yf(\boldsymbol{X}))]$.

Theorem 1 shows that for any fixed $\varepsilon$, the minimizer of (4) is the same as the Bayes rule [14].

**Theorem 1.** *Under Assumption 1, for any fixed $\varepsilon \geq 0$, function*

$$f^*(\boldsymbol{x}) = \begin{cases} 1 + \varepsilon, & \eta(\boldsymbol{x}) > t_- \\ \varepsilon \cdot sign(\eta(\boldsymbol{x}) - \frac{1}{2}), & t_+ \leq \eta(\boldsymbol{x}) \leq t_- \\ -(1 + \varepsilon), & f(\boldsymbol{x}) < t_+ \end{cases}.$$

*is the minimizer to (4) and a minimizer to (2).*

A key result in many machine learning literature (such as [3], [30] or [2]) was that the excess risk of 0-1 classification loss is bounded by the excess risk of surrogate loss. Here we show a similar result for the confidence set problem. That is, the excess ambiguity $R(f, \varepsilon) - R(f^*, \varepsilon)$ vanishes as $R_H(f, \varepsilon) - R_H(f^*, \varepsilon)$ goes to 0.

**Theorem 2.** *Under Assumption (2), for any $\varepsilon \geq 0$, and $\forall f$ satisfying the constraints in (2), there exists $C' = \frac{1}{4c^2} + \frac{1}{2c} > 0$ such that the following inequality holds,*

$$C'(R_H(f, \varepsilon) - R_H(f^*, \varepsilon)) \geq R(f) - R(f^*).$$

Note that $C'$ does not depend on $\varepsilon$.

## 4.2 Finite-sample properties

Denote the Reproducing Kernel Hilbert Space (RKHS) with bounded norm as $\mathcal{H}_K(s) = \{f : \mathcal{X} \to \mathbb{R} | f(\boldsymbol{x}) = h(\boldsymbol{x}) + b, h \in \mathcal{H}_K, ||h||_{\mathcal{H}_K} \leq s, b \in \mathbb{R}\}$ and $r = \sup_{\boldsymbol{x} \in \mathcal{X}} K(\boldsymbol{x}, \boldsymbol{x})$. For a fixed $\varepsilon$, define the space of constrained discriminant functions as $\mathcal{F}_\varepsilon((\alpha_{-1}, \alpha_1)) = \{f : \mathcal{X} \to \mathbb{R} | E(H_{-\varepsilon}(Yf(\boldsymbol{X}))|Y = j) \leq \alpha_j, j = \pm 1\}$, and its empirical counterpart as $\hat{\mathcal{F}}_\varepsilon((\alpha_-, \alpha_+)) = \{f : \mathcal{X} \to \mathbb{R} | n_j^{-1} \sum_{i:y_i=j} H_{-\varepsilon}(y_i f(\boldsymbol{x}_i)) \leq \alpha_j, j = \pm 1\}$. Moreover, we define the feasible function space $\mathcal{F}_\varepsilon(\kappa, s) = \mathcal{H}_K(s) \cap \mathcal{F}_\varepsilon((\alpha_{-1} - \frac{\kappa}{\sqrt{n_{-1}}}, \alpha_1 - \frac{\kappa}{\sqrt{n_1}}))$ and its empirical counterpart $\hat{\mathcal{F}}_\varepsilon(\kappa, s) = \mathcal{H}_K(s) \cap \hat{\mathcal{F}}_\varepsilon((\alpha_{-1} - \frac{\kappa}{\sqrt{n_{-1}}}, \alpha_1 - \frac{\kappa}{\sqrt{n_1}}))$. Lastly, consider a subset of the Cartesian product of the above feasible function space and the space for $\varepsilon$, $\mathcal{F}(\kappa, s) = \{(f, \varepsilon), f \in \mathcal{F}_\varepsilon(\kappa, s), \varepsilon \geq 0\}$ and its empirical counterpart $\hat{\mathcal{F}}(\kappa, s) = \{(f, \varepsilon), f \in \hat{\mathcal{F}}_\varepsilon(\kappa, s), \varepsilon \geq 0\}$. Then optimization problem (3) of our proposed method can be written as

$$\min_{(f,\varepsilon) \in \hat{\mathcal{F}}(0,s)} \frac{1}{n} \sum_{i=1}^n H_\varepsilon(y_i f(\boldsymbol{x}_i)). \tag{5}$$

In Theorem 3, we give the finite-sample upper bound for the non-coverage rate.

**Theorem 3.** *Let $(f, \varepsilon)$ be a solution to optimization problem (5), then with probability at least $1 - 2\zeta$, $Z = \sqrt{sr}/\sqrt{n}$, $T_n(\zeta) = \{2sr \log(1/\zeta)/n\}^{1/2}$ and $r = \sup_{\mathcal{X}} K(\boldsymbol{x}, \boldsymbol{x})$*

$$P_j(Yf(\boldsymbol{X}) < -\varepsilon) \leq E[H_{-\varepsilon}(Yf(\boldsymbol{X}))|Y = j] \leq \frac{1}{n_j} \sum_{y_i = j} H_{-\varepsilon}(y_i f(\boldsymbol{x}_i)) + 3T_{n_j}(\zeta) + Z(n_j).$$

Theorem 3 suggests that if we want to control the non-coverage rate on average at the nominal $\alpha_{-1}$ or $\alpha_1$ rates with high probability, we should choose the $\alpha_{-1}$ or $\alpha_1$ values to be slightly smaller than the desired ones in optimization (3) in practice. In particular, we need to make $\frac{1}{n_j} \sum_{y_i=j} H_{-\varepsilon}(y_i f(\boldsymbol{x}_i)) + 3T_{n_j}(\zeta) + Z(n_j) \leq \alpha_j$. Note that the remainder terms $3T_{n_j}(\zeta) + Z(n_j)$ will vanish as $n_{-1}, n_1 \to \infty$.

The next theorem ensures that the empirical ambiguity probability from solving (5) based on a finite sample will converge to the ambiguity given by the solution on an infinite sample (under the constraints $E(H_{-\varepsilon}(Yf(\boldsymbol{X}))|Y = j) \leq \alpha_j, j = \pm 1$).

**Theorem 4.** *Let $(\hat{f}, \hat{\varepsilon})$ be the solution of the optimization problem (6)*

$$\min_{(f,\varepsilon) \in \hat{\mathcal{F}}(\kappa,s)} \frac{1}{n} \sum_{i=1}^n H_\varepsilon(y_i f(\boldsymbol{x}_i)). \tag{6}$$

*with $\kappa = (6 \log(\frac{1}{\zeta}) + 1)\sqrt{sr}$. Then with probability $1 - 6\zeta$, and large enough $n_{-1}$ and $n_1$ we have*
*(i). $\hat{f} \in \mathcal{F}_{\hat{\varepsilon}}(0, s)$, and*
*(ii). $R_H(\hat{f}, \hat{\varepsilon}) - \min_{(f,\varepsilon) \in \mathcal{F}(0,s)} R_H(f, \hat{\varepsilon}) \leq \kappa(2n^{-1/2} + 4 \min\{\alpha_{-1}, \alpha_1\}^{-1} \min\{\sqrt{n_{-1}}, \sqrt{n_1}\}^{-1}).$*

In our study we analyze formula (5) where $J(f)$ appears in the constraint instead of the regularized formula (3) for technical convenience. This comes at a price of a fixed upper bound $s$ on $J(f)$. We can revise the statements of Theorems 3 and 4 so that $s$ increases with $n$ to infinity (with a price of a slower convergence rate.) It is possible to derive the results for the regularized version based on (3). Since at the optimality it is easy to show that $J(f) \leq 2/\lambda$ (this is done by showing that the objective is at most 2, when $f \equiv 0$ and $\varepsilon = 1$,) we may rewrite $s$ in Theorem 3 in terms of $\lambda$.

## 5 Algorithms

In this section, we give details of the algorithm. Similar to the SVM implementation, we propose to solve the dual problem. We start with the linear SVM with $L_2$ norm for illustrative purposes. After introducing two sets of slack variables, $\eta_i = (1 - \varepsilon - y_i(\boldsymbol{x_i}^T \boldsymbol{\beta} + b))_+$ and $\xi_i = (1 + \varepsilon - y_i(\boldsymbol{x_i}^T \boldsymbol{\beta} + b))_+$, we can show that (3) is equivalent to (7),

$$\min_{\Theta} \frac{1}{2} ||\boldsymbol{\beta}||_2^2 + \lambda' \sum_i^n \xi_i \tag{7}$$

$$\text{subject to } y_i(\boldsymbol{x_i}^T\boldsymbol{\beta} + b) \geq 1 + \varepsilon - \xi_i, \quad y_i(\boldsymbol{x_i}^T\boldsymbol{\beta} + b) \geq 1 - \varepsilon - \eta_i \quad \text{for all } i = 1, 2, ..., n,$$

$$\xi_i \geq 0, \quad \sum_{y_i=-1} w_i\eta_i \leq n_{-1}\alpha_{-1}, \quad \eta_i \geq 0, \quad \sum_{y_i=1} w_i\eta_i \leq n_1\alpha_1, \quad \varepsilon \geq 0.$$

Here $\Theta$ is the collection of all variables of interest, namely $\Theta = \{\varepsilon, \boldsymbol{\beta}, b, \{\xi_i\}_{i=1}^n, \{\eta_i\}_{i=1}^n\}$. We can then solve it via the quadratic programming below,

$$\min_{\Theta'} \frac{1}{2}\sum_{i=1}^n\sum_{j=1}^n (\zeta_i + \tau_i)(\zeta_j + \tau_j)y_iy_j\boldsymbol{x}_i'\boldsymbol{x}_j - \sum_{i=1}^n \zeta_i - \sum_{i=1}^n \tau_i + n_{-1}\alpha_{-1}\theta_{-1} + n_1\alpha_1\theta_1 \quad (8)$$

$$\text{subject to } 0 \leq \zeta_i \leq \lambda', \quad 0 \leq \tau_i \leq \theta_{y_i}w_i, \quad \sum_{i=1}^n \zeta_iy_i + \sum_{i=1}^n \tau_iy_i = 0, \quad \sum_{i=1}^n \zeta_i - \sum_{i=1}^n \tau_i \geq 0.$$

Here $\Theta' = \{\{\zeta_i\}_{i=1}^n, \{\tau_i\}_{i=1}^n, \theta_{-1}, \theta_1\}$ consists of all the variables in the dual problem. The above optimization may be solved by any efficient quadratic programming routine. After solving the dual problem, we can find $\boldsymbol{\beta}$ by $\boldsymbol{\beta} = \sum_i^n \zeta_iy_i\boldsymbol{x_i} + \sum_i^n \tau_iy_i\boldsymbol{x_i}$. Then we can plug $\boldsymbol{\beta}$ into the primal problem and find $b$ and $\varepsilon$ by linear programming.

For nonlinear $f$, we can adopt the widely used 'kernel trick'. Assume $f$ belongs to a Reproducing Kernel Hilbert Space (RKHS) with a positive definite kernel $K$, $f(\boldsymbol{x}) = \sum_{i=1}^n c_iK(\boldsymbol{x}_i, \boldsymbol{x}) + b$. In this case the dual problem is the same as above except that $\boldsymbol{x}_i'\boldsymbol{x}_j$ is replaced by $K(\boldsymbol{x}_i, \boldsymbol{x}_j)$. After the solution has been found, we then have $c_i = \zeta_i + \tau_i$. Common choices for the kernel function includes the Gaussian kernel and the polynomial kernel.

## 6 Numerical Studies

In this section, we compare our confidence-support vector machine (CSVM) method and methods based on the plug-in principal, including $L_2$ penalized logistic regression [12], kernel logistic regression [31], kNN [1], random forest [15] and SVM [5] using both simulated and real data.

In the study, we use solver `Cplex` to solve the quadratic programming problem arising in CSVM. For other methods, we use existing R packages `glmnet`, `gelnet`, `class`, `randomForest` and `e1071`.

### 6.1 Simulation

We study the numerical performance over a large variety of sample sizes. In each case, an independent tuning set with the same sample size as the training set is generated for parameter tuning. The testing set has 20000 observations (10000 or nearly 10000 for each class). We run the simulation multiple times (1,000 times for Example 1 and 100 times for Example 2 and 3) and report the average and standard error. Both non-coverage rates are set to 0.05.

We select the best parameter $\lambda$ and the hyper-parameter for kernel methods as follows. We search for the optimal $\rho$ in the Gaussian kernel $\exp\left(-\|x - y\|^2/\rho^2\right)$ from the grid $10^{\{-0.5, -0.25, 0, 0.25, 0.5, 0.75, 1\}}$ and the optimal degree for polynomial kernel from $\{2, 3, 4\}$. For each fixed candidate hyper-parameter, we choose $\lambda$ from a grid of candidate values ranging from $10^{-4}$ to $10^2$ by the following two-step searching scheme. We first do a rough search with a larger stride $\{10^{-4}, 10^{-3.5}, \ldots, 10^2\}$ and get the best parameter $\lambda_1$. Then we do a fine search from $\lambda_1 \times \{10^{-0.5}, 10^{-.4}, \ldots, 10^{0.5}\}$. After that, we choose the optimal pair which gives the smallest tuning ambiguity and has the two non-coverage rates for the tuning set controlled.

To adapt traditional classification methods to the confidence set learning problem, we use the plug-in principal [14]. To improve the performance, we make use of the suggested robust implementation in [14] for all the methods. Specifically, we first obtain an estimate of $\eta$ (such as by logistic regression, kernel logistic regression, $k$NN and random forest) or a monotone proxy of it (such as the discriminant function $f$ in CSVM and SVM), then choose thresholds $\hat{t}_{-1}$ and $\hat{t}_1$ which are two sample quantiles of $\widehat{\eta}(\boldsymbol{x})$ (or $f(\boldsymbol{x})$) among the tuning set so that the non-coverage rates for the tuning set match the nominal rates. The final predicted sets are induced by thresholding $\widehat{\eta}(\boldsymbol{x})$ (or $f(\boldsymbol{x})$) using $\hat{t}_{-1}$ and $\hat{t}_1$.

Because there are two non-coverage rates and one ambiguity size to compare here, how to make fair comparison becomes a tricky problem since one classifier can sacrifice the non-coverage rate to gain

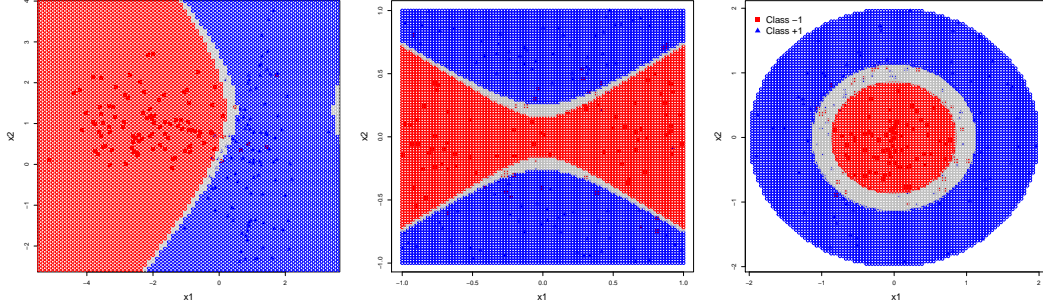

Figure 2: Scatter plots of the first two dimensions for the simulated data with Bayes rules showing the two definite regions and the ambiguity region.

in ambiguity. One by-product of the robust implementation above is that the non-coverage rate of most of the methods will become very similar and we only need to compare the size of the ambiguity.

We also include a naive SVM approach ('SVM_r' in plots below) whose discriminant function is obtained in the traditional way, but which induces confidence sets by thresholding in the same way described above.

We consider three different simulation scenarios. In the first scenario we compare the linear approaches (SVM and penalized logistic regression), while in the next two cases we consider nonlinear methods. In all cases, we add additional noise dimensions to the data. These noise covariates are normally distributed with mean $\mathbf{0}$ and $\Sigma = \text{diag}(1/p)$, where $p$ is the total dimension of the data.

**Example 1 (Linear model with nonlinear Bayes rule):** In this scenario, we have two normally distributed classes with different covariance matrices. In particular, denote $X|Y = j \sim \mathcal{N}(\mu_j, \Sigma_j)$ for $j = \pm 1$, then $\mu_{-1} = (-2, 1)^T$, $\mu_1 = (1, 0)^T$, and $\Sigma_{-1} = \text{diag}(2, \frac{1}{2})$, $\Sigma_1 = \text{diag}(\frac{1}{2}, 2)$. The prior probabilities of both classes are the same. Lastly, we add eight dimensions of noise covariates to the data. The data are illustrated in the left penal of Figure 2. We compare linear CSVM, and the plug-in methods $L_2$ penalized logistic regression [8] and naive linear SVM to estimate $\eta$.

**Example 2 (Moderate dimensional polynomial boundary):** This case is similar to the one in [29]. First we generate $x_1 \sim \text{Unif}[-1, 1]$ and $x_2 \sim \text{Unif}[-1, 1]$. Define functions $f_j(\boldsymbol{x}) = j(-3.6x_1^2 + 7.2x_2^2 - 0.8)$, $j = \pm 1$. Then we set $\eta(\boldsymbol{x}) = f_1(\boldsymbol{x})/(f_{-1}(\boldsymbol{x}) + f_1(\boldsymbol{x}))$, where $\boldsymbol{x} = (x_1, x_2)$. We then add 98 covariates on top of the 2-dimensional signal. The data are illustrated in the middle penal of Figure 2. In this scenario, we choose to use the polynomial kernel for all the kernel based methods.

**Example 3 (High-dimensional donut):** We first generate a two-dimensional data, $(r_i, \theta_i)$ where $\theta_i \sim \text{Unif}[0, 2\pi]$, $r_i|(Y = -1) \sim \text{Uniform}[0, 1.2]$, and $r_i|(Y = +1) \sim \text{Unif}[0.8, 2]$. Then we define the two-dimensional $\boldsymbol{X}_i = (r_i \cos(\theta_i), r_i \sin(\theta_i))$. The data are illustrated in the right penal of Figure 2. We then add 498 covariates on top of the 2-dimensional signal. We use the Gaussian kernel, $K(x, y; \rho) = \exp\left(-\|x - y\|^2/\rho^2\right)$ for all the kernel based methods.

Our methods are improved using the robust implementation. The results are reported in Figure 3. We also show the performance of CSVM with weighting but without robust implementation. For Example 1, our CSVM method gives a significantly smaller ambiguity than either logistic regression or naive SVM. In Example 2 and Example 3, our method gives a smaller or at least comparable ambiguity to the best plug-in method, which is kernel logistic regression. Our weighted CSVM performs the best when sample size is small in the linear case and it outperforms kNN, Random Forest and naive SVM in nonlinear cases. It is not surprising that the naive SVM method performs significantly worse than all other methods in the nonlinear settings, as the hinge loss is well known to not lead to consistent estimates for class probabilities (see [18]). The non-coverage rates (not shown here) of CSVM, random forest, kernel logistic regression and naive SVM methods are close to each other while CSVM without robust implementation and kNN have similar non-coverage rates. A detailed comparison can be found in the Supplementary Material.

## 6.2 Real Data Analysis

We conduct the comparison on the hand-written zip code data [13]. The data set consists of many $16 \times 16$ pixel images of handwritten digits. It is widely used in the classification literature. There are

both training and testing sets defined in it. Lei [14] used the same dataset for illustrating the plug-in methods. We choose this dataset to directly compare with the plug-in methods.

Following Lei [14], to form a binary classification problem, we use the subset of the data containing digits $\{0, 6, 8, 9\}$. Images with digits 0, 6, 9 are labeled as class $-1$ (they are digits with one circle) and those with digit 8 (two circles) are labeled as class $+1$. Previous studies [21] pointed out that there was discrepancies between the training and testing set of this data set. So in this study we first mixed the training and testing data and then randomly split into new training, tuning and testing data. The training and tuning data both have sample size 800, with 600 from class $-1$ and 200 from class 1 to preserve the unbalance nature of the data set. During training, we oversample class 1 by counting each observation three times to alleviate the unbalanced classes issue.

Although Lei [14] set both nominal non-coverage rates to be 0.05 in their study which focused on linear methods, it needs to be pointed out that many nonlinear classifiers, such as SVM with Gaussian kernel, can achieve this non-coverage rate without introducing any ambiguity. Therefore we reduce the non-coverage rate to 0.01 for both classes to make the task more challenging.

We apply Gaussian kernel for CSVM, and compare with kernel logistic regression with Gaussian kernel, random forest, kNN and naive SVM with Gaussian kernel on this data set.

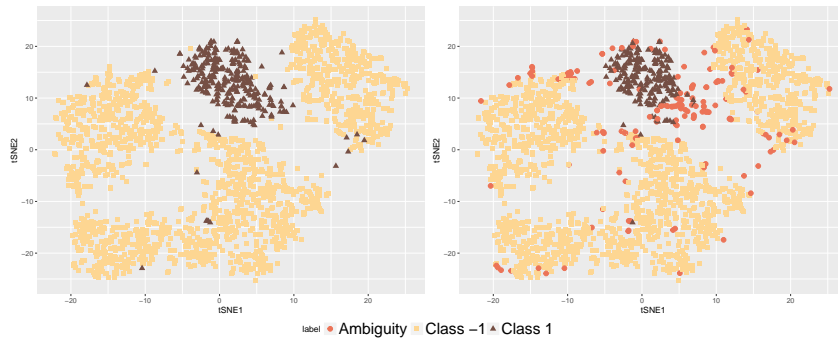

Figure 4: An illustration of CSVM method using t-SNE. The left penal shows the true labels, and the right panel the predicted label for weighted CSVM.

The results are summarized in Table 1 with numbers in percentage. CSVM gives better results than all the plug-in methods. We plot the zip code data using t-distributed stochastic neighbor embedding (t-SNE) [17] to give a visualization of our method and the data.

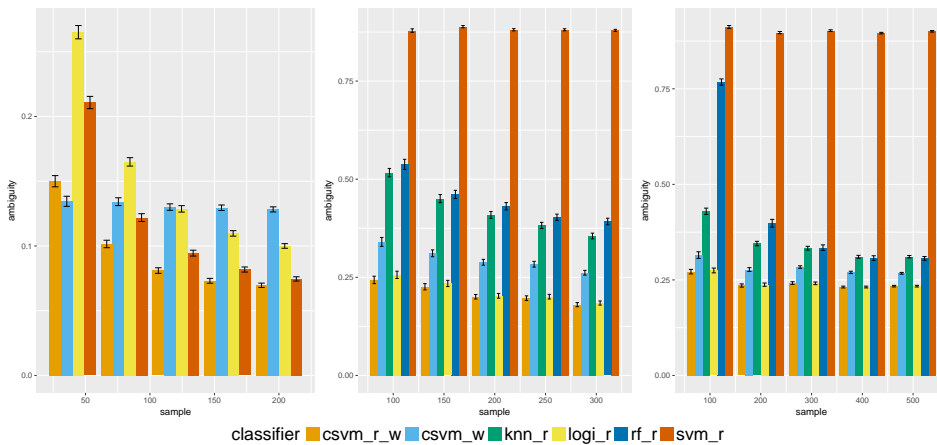

Figure 3: Outcome of ambiguities in three simulation settings. Non-coverage rates are similar among different methods and are not shown here. CSVM has the smallest ambiguity.

| Classifier | CSVM | CSVM(r) | KNN(r) | KLR(r) | RF(r) | naive SVM(r) |
|---|---|---|---|---|---|---|
| Non-coverage(-1) | 0.05(0.005) | 1.02(0.05) | 0.81(0.04) | 0.98(0.05) | 0.95(0.04) | 1.00(0.05) |
| Non-coverage(+1) | 0.56(0.06) | 1.19(0.11) | 1.04(0.09) | 1.25(0.10) | 1.10(0.11) | 1.27(0.11) |
| Ambiguity | 8.29(0.18) | 2.52(0.13) | 10.21(2.12) | 3.46(0.17) | 7.55(0.37) | 2.66(0.13) |

Table 1: CSVM gives better or comparable outcome to the best plug-in method.

It can be seen that the ambiguity region mainly lies on the boundary between the two classes. In particular, they cover those points which appear to be closer to the class other than the one they really belong to. Moreover, it can be seen that the union of the ambiguity region and the predicted region for either class, covers almost all the ground of that class (defined by the true labels). This is not surprising since the non-coverage rate of CSVM is set to be a small number of 1% in this case.

## 7  Conclusion and future works

In this work, we propose to learn confidence sets using support vector machine. Instead of a plug-in approach, we use empirical risk minimization to train the classifier. Theoretical studies have shown the effectiveness of our approach in controlling the non-coverage rate and minimizing the ambiguity.

We make use of many well understood advantages of SVM to solve the problem. For instance the 'kernel trick' allows more flexibility and empowers us to conduct classification in nonlinear cases.

Hinge loss function is not the only surrogate loss that can be used. There are many other useful loss functions with good properties in different scenarios [16].

Confidence set learning for multi-class case is also an interesting future work. This has a natural connection to the literature of multi-class classification with confidence [20], classification with reject and refine options [28] and conformal learning [21].

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
