[Supplementary Material]

# Supplementary Material of "Learning Confidence Sets using Support Vector Machines"

Wenbo Wang, Xingye Qiao*

November 28, 2018

## 1 Proof

This part will give proofs to some of the statements and theorems in the main part.

**Proof of Dual Representation**

Firstly, the Lagrangian of problem (6) is

$$
\begin{aligned}
L(\beta, b, \{\xi_i\}_{i=1}^n, \{\eta_i\}_{i=1}^n, \varepsilon) = &\frac{1}{2}||\beta||_2^2 + \lambda' \sum_{i=1}^n \xi_i + \sum_{i=1}^n \zeta_i(1 + \varepsilon - \xi_i - y_i(\boldsymbol{x_i}^T\beta + b)) \\
&+ \sum_{i=1}^n \tau_i(1 - \varepsilon - \eta_i - y_i(\boldsymbol{x_i}^T\beta + b)) - \sum_{i=1}^n \rho_i\xi_i - \sum_{i=1}^n \gamma_i\eta_i \\
&+ \theta_{-1}(\sum_{y_i=-1} w_i\eta_i - n_{-1}\alpha_{-1}) + \theta_1(\sum_{y_i=1} w_i\eta_i - n_1\alpha_1) - \nu\varepsilon. \quad (1)
\end{aligned}
$$

Then we consider the Karush-Kuhn-Tucker conditions. We write $L(\beta, b, \varepsilon, \{\xi_i\}_{i=1}^n, \{\eta_i\}_{i=1}^n)$ as $L$ for simplicity.

$$
\frac{\partial L}{\partial \beta} = \beta - \sum_i^n \zeta_i y_i \boldsymbol{x_i} - \sum_i^n \tau_i y_i \boldsymbol{x_i} = 0,
$$

$$
\frac{\partial L}{\partial b} = -\sum_{i=1}^n \zeta_i y_i - \sum_{i=1}^n \tau_i y_i = 0,
$$

$$
\frac{\partial L}{\partial \xi_i} = \lambda - \zeta_i - \rho_i = 0 \quad \text{for} \quad \forall i,
$$

$$
\frac{\partial L}{\partial \eta_i} = -\tau_i - \gamma_i + w_i\theta_{-1} = 0 \quad \text{for} \quad y_i = -1,
$$

$$\frac{\partial L}{\partial \eta_i} = -\tau_i - \gamma_i + w_i \theta_1 = 0 \quad \text{for} \quad y_i = 1,$$

$$\frac{\partial L}{\partial \varepsilon} = \sum_i^n \zeta_i - \sum_i^n \tau_i - \nu = 0,$$

$$\zeta_i (1 + \varepsilon - \xi_i - y_i(\boldsymbol{x_i}^T \beta - b)) = 0 \quad \text{for} \quad i = 1, 2, ..., n,$$

$$\tau_i (1 - \varepsilon - \eta_i - y_i(\boldsymbol{x_i}^T \beta - b)) = 0 \quad \text{for} \quad i = 1, 2, ..., n,$$

$$\rho_i \xi_i = 0 \quad \text{for} \quad i = 1, 2, ..., n,$$

$$\gamma_i \eta_i = 0 \quad \text{for} \quad i = 1, 2, ..., n,$$

$$\theta_{-1} \left( \sum_{y_i=-1} \eta_i - n_{-1} \alpha_{-1} \right) = 0,$$

$$\theta_1 \left( \sum_{y_i=1} \eta_i - n_1 \alpha_1 \right) = 0,$$

$$\nu \varepsilon = 0.$$

After plugging the KKT conditions into expression 1, we can get the dual problem. □

## Proof of Theorem 1

In order to prove Theorem 1 and 2, we need to first introduce another risk function, $\bar{R}(f, \varepsilon) = P(Yf(X) < \varepsilon) + \frac{1}{2}P(|f(X)| \leq \varepsilon)$ and the optimization problem associate with it.

$$\min \ \bar{R}(f, \varepsilon) \tag{2}$$
$$\text{subject to} \quad P_j(Yf(X) < -\varepsilon) \leq \alpha_j, \quad j = \pm 1.$$

And here is a Lemma come with it.

**Lemma 1.** *Under Assumption 1, for any fixed $\varepsilon \geq 0$, the discrimination function $f^*$ such that*

$$f^*(x) = \begin{cases} 1 + \varepsilon, & \eta(x) > t_{-1} \\ \varepsilon * sgn(\eta(x) - \frac{1}{2}), & t_1 \leq \eta(x) \leq t_{-1} \\ -(1 + \varepsilon), & f(x) < t_1 \end{cases}$$

*is a solution to the optimization problem (2) (in the main work) and (2).*

Denote $C_{-1}^* = \{x : f^*(x) \leq \varepsilon\}$ and $C_1^* = \{x : f^*(x) \geq -\varepsilon\}$ and the set classifier introduced by $f^*$ is $\phi^*$. The Assumption 1 ensures that $C_{-1}^* \cup C_1^* = \mathcal{X}$. Let's denote $C_{-1}^* \cap C_1^*$ by $C_0^*$. The first part of this lemma is of vital importance in showing the Fisher consistency of the proposed method and the second part will be used in proving Theorem 2.

The optimality of $\phi^*$ to Problem (2) (in the main work) is proved in Lei (2014). Here we prove the optimality of $f^*$ for problem 2. The technique used in the following prove is fairly straight-forward in statistical decision and game theory. We start the proof with looking for a so-called complete set of $f$. After that, we only need to focus on this set of discriminant functions. We firstly make two definitions to simplify our proof.

**Definition 1.** *For any (inequality) constrained optimization problem with m constrains*

$$\min L(f)$$
$$\text{subject to } C_i(f) \leq c_i \quad i = 1, ..., m,$$

*two function $f_1$ and $f_2$, $f_1$ is said to be as good as than $f_2$ when $L(f_1) \leq L(f_2)$ and $C_i(f_1) \leq C_i(f_2)$ for $\forall i$, and better than $f_2$ when one of those inequality holds strictly.*

**Definition 2.** *Given the distribution of $\boldsymbol{X}$ and $Y$, denoted as $P$. Define a class of function, $\mathcal{F}^*(a_1, a_2; b_1, b_2)$ consists all functions $f$ which take at most two distinct non-negative values $a_1 < a_2$ for $\{x : \eta > \frac{1}{2}\}$ and at most two distinct non-positive values $b_1 > b_2$ for $\{x : \eta(x) < \frac{1}{2}\}$. A constrained optimization problem $\mathcal{O}$ is said to be simple monotone with respect to $\mathcal{F}^*(a_1, a_2, b_1, b_2)$, if it satisfies:*

*(i). $\mathcal{F}^*(a_1, a_2, b_1, b_2)$ is a complete class of the problem, which means $\forall f$, $\exists g \in \mathcal{F}^*$, and $g$ is as good as $f$.*

*(ii). If there exist disjoint $B_1, B_2 \in \mathcal{X}$, such that $P_{-1}(B_1) = P_{-1}(B_2) > 0$, for $\forall x_1 \in B_1, x_2 \in B_2$, $\eta(x_1) > \eta(x_2) > \frac{1}{2}$. Moreover, for any pairs of function in $\mathcal{F}^*$, $f_1(x)$ and $f_2(x)$ such that $f_1(x) = f_2(x)$ for $\forall x \notin B_1 \cup B_2$, and $f_1(x) = \begin{cases} a_1, & \forall x \in B_1 \\ a_2, & \forall x \in B_2 \end{cases}$ and $f_2(x) = \begin{cases} a_2, & \forall x \in B_1 \\ a_1, & \forall x \in B_2 \end{cases}$, $f_2$ is better than $f_1$.*

*(iii). If there exist disjoint $B_1', B_2' \in \mathcal{X}$, such that $P_1(B_1') = P_1(B_2') > 0$, for $\forall x_1 \in B_1', x_2 \in B_2'$, $\eta(x_1) < \eta(x_2) < \frac{1}{2}$. Moreover, for any pairs of function in $\mathcal{F}^*$, $f_1(x)$ and $f_2(x)$ such that $f_1(x) = f_2(x)$ for $\forall x \notin B_1' \cup B_2'$, and $f_1(x) = \begin{cases} b_1, & \forall x \in B_1' \\ b_2, & \forall x \in B_2' \end{cases}$ and $f_2(x) = \begin{cases} b_2, & \forall x \in B_1' \\ b_1, & \forall x \in B_2' \end{cases}$, $f_2$ is better than $f_1$.*

It can be shown that a complete class of a simple monotone optimization problem can be astonishingly simple. We are going to further shrink our focus to the functions which only depend on $\eta(x)$ rather than $x$. In other words, we can regard $\eta$ as a sufficient statistic of $x$.

**Lemma 2.** *If an optimization $\mathcal{O}$ is simple monotone with respect to $\mathcal{F}^*(a_1, a_2; b_1, b_2)$, then a solution of $\mathcal{O}$ in $\mathcal{F}^*$ takes the form*

$$f(x) = \begin{cases} a_2, & \eta(x) > t \\ a_1, & \frac{1}{2} \leq \eta(x) \leq t \\ b_1, & t' \leq \eta(x) < \frac{1}{2} \\ b_2, & \eta(x) < t' \end{cases}$$

*for some $t' \leq \frac{1}{2} \leq t$ almost surely.*

*Proof.* Let's prove there exists a $\frac{1}{2} \leq t \leq 1$ such that $f(x) = a_1, \forall x$, such that $\frac{1}{2} < \eta(x) < t, a.s.$ and $f(x) = a_2, \forall \eta(x) > t, a.s..$ Define $T_1 = \{t : \exists C, P(C) > 0, \eta(x) > t, f(x) = a_1, \forall x \in C\}$ and $T_2 = \{t : \exists C', P(C') > 0, \eta(x) < t, f(x) = a_2, \forall x \in C'\}$. Firstly, if $T_1 = \emptyset$,

then $t = \frac{1}{2}$ and similarly, if $T_2 = \emptyset$, then $t = 1$. So now we can assume $T_1$ and $T_2$ are nonempty.

If $t_1 \in T_1$, then by definition $t_2 \in T_1, \forall t_2 < t_1$, so that $T_1$ is a interval and $\frac{1}{2} \in T_1$. Similarly, $T_2$ is also a interval and $1 \in T_2$. Moreover, $T_1$ and $T_2$ are open interval in $[\frac{1}{2}, 1]$. By definition, if $t_1 \in T_1$, then we have $P(C \cap (\cup_n^\infty \{x : \eta(x) > t_1 + \frac{1}{n}\})) > 0$, so $\exists m$, such that $P(C \cap \{x : \eta(x) > t_1 + \frac{1}{m}\}) > 0$. Thus we have $t_1 + \frac{1}{m} \in T_1$ as well. If $T_1 \cap T_2 \neq \emptyset$, then we have a $t' \in T_1 \cap T_2$, which indicates there exists $\forall C_1, C_2 \in \mathcal{X}$ such that $P(C_1), P(C_2) > 0$, $1 > \eta(x_1) > \eta(x_2) > \frac{1}{2}$, and $\hat{f}(x_1) = a_2$, $\hat{f}(x_1) = a_2$, $\forall x_1 \in C_1, x_2 \in C_2$. If we have $P(C_1), P(C_2) > 0$. Then we can choose two subsets of $C_1$ and $C_2$, named $C_1'$ and $C_2'$, such that $P_{-1}(C_1') = P_{-1}(C_2') > 0$, because $P_{-1}$ is continuous with positive density and $\eta$ is continuous ($P(\eta = 1) = 0$). This will leads to a contradiction with the (ii) of 2. If $T_1 \cap T_2 = \emptyset$, then we can choose a point t in $[\sup\{T_1\}, \inf\{T_2\}]$ and it will satisfy our purpose.

By similar argument, we can show there exists a $0 \leq t' \leq \frac{1}{2}$.

$\square$

**Proof of Lemma 1**: We want to show that 2 is simple monotone with respect to $\mathcal{F}^*(\varepsilon + 1, 0, 0, -(\varepsilon + 1))$.

Because optimization problem 2 can be regarded as an optimization problem for classifiers, it is sufficient to consider functions $f$ with 3 values, $\varepsilon + 1$, $-(\varepsilon + 1)$, 0, that is $f \in \mathcal{F}(\varepsilon) := \{f : \mathcal{X} \to \{\varepsilon + 1, -(\varepsilon + 1), 0\}\}$.

Firstly, we need to prove $sign(\eta(X) - \frac{1}{2})\hat{f}(X) \geq 0$ with probability 1 for any $\hat{f}$, a solution of 2 in $\mathcal{F}(\varepsilon)$. If there is a set $A \subset \mathcal{X}$, $\forall x \in A$, $\eta(x) > \frac{1}{2}$, $\hat{f} = -(\varepsilon + 1)$ and $P(A) > 0$, then we can consider another function $f_A$ such that $f_A(x) = \hat{f}(x), \forall x \in A^c$ but $f_A(x) = 0, \forall x \in A$. $f_A$ will be better than $\hat{f}$. It is easy to check that two constraints still hold for $f_A$. But the objective function will be smaller, because $\frac{1}{2}P(|\hat{f}(X)| \leq \varepsilon) + P(Y\hat{f}(X) < -\varepsilon) - \frac{1}{2}P(|f_A(X)| \leq \varepsilon) + P(Yf_A(X) < -\varepsilon) = \frac{1}{2}P(|\hat{f}(X)| \leq \varepsilon, X \in A) + P(Y\hat{f}(X) < -\varepsilon, X \in A) - \frac{1}{2}P(|f_A(X)| \leq \varepsilon, X \in A) + P(Yf_A(X) < -\varepsilon, X \in A) = E(\eta(X)1_{X \in A}) > 0$. This will lead to a contradiction with the optimality of $\hat{f}$.

We only give the proof for part (ii) in 2, and part (iii) can be proved analogously. We can check the constraints and objective function one by one.

Firstly, $P_1(Yf_1(X) < -\varepsilon) = P_1(Yf_2(X) < -\varepsilon)$ because the set in which $f_1$ and $f_2$ take $-(1 + \varepsilon)$ are the same. Secondly, $P_{-1}(Yf_1(X) < -\varepsilon) - P_{-1}(Yf_2(X) < -\varepsilon) = P_{-1}(B_2) - P_{-1}(B_1) = 0$. Lastly, $\frac{1}{2}P(|f_2(X)| \leq \varepsilon) + P(Yf_2(X) < -\varepsilon) - (\frac{1}{2}P(|f_1(X)| \leq \varepsilon) + P(Yf_1(X) < -\varepsilon)) = E(1_{(X \in B_2)}(\eta(X) - \frac{1}{2})) - E(1_{(X \in B_1)}(\eta(X) - \frac{1}{2})) < 0$. This comes from the fact that $\eta(x_1) - \frac{1}{2} > \eta(x_2) - \frac{1}{2} > 0, \forall x_1 \in B_1, x_2 \in B_2$ and $P(B_2) < P(B_1)$. The last inequality come from $P(Y = 1|X \in B_1) > P(Y = 1|X \in B_2)$ and $P_{-1}(B_1) = P_{-1}(B_2)$.

Then by Lemma 2, we can see that the solution of 2 only depends on $\eta$.

The next part of this proof is to find out the optimal $t$ and $t'$. Let's show that the optimal choice of $t$ is $t_{-1}$. If $t \neq t_{-1}$ for $\hat{f}$, then $t > t_{-1}$ and $P(\eta(x) \leq t) < 1 - \alpha_{-1}$ (note that $\eta$ is continuous a.s.), otherwise $\hat{f}$ does not satisfy the constraint that $P_{-1}(Yf(X) < -\varepsilon) \leq \alpha_{-1}$. Then If we consider another function $\hat{f}^*$ such that $\hat{f}'(x) = 0, \forall x, s.t. \quad t_{-1} < \eta(x) < t$ and $\hat{f}^*(x) = \hat{f}(x)$ elsewhere. Denote $C' = \{x : t_{-1} < \eta(x) < t\}$ and $P(C') > 0$. Then we

have that $\frac{1}{2}P(|\hat{f}(X)| \le \varepsilon) + P(Y\hat{f}(X) < -\varepsilon) - (\frac{1}{2}P(|\hat{f}'(X)| \le \varepsilon) + P(Y\hat{f}'(X) < -\varepsilon)) = E((\frac{1}{2} - (1 - \eta(X)))\mathbb{1}\{(C')\}) > 0$. So $t = t_{-1}$. The optimal choice for $t'$ can be found in a similar way.

The proof is completed by observing $f^*$ gives exactly the same $\bar{R}$ loss.

$\square$

Now let's start to prove Theorem 1.

The argument in the proof is similar to Lemma 1. We are going to show the optimization problem (9) (in the main work) is simply monotone. We consider our proof in two parts. The first is to show the minimizer of optimization problem (4) (in the main work) can only takes four values and is Fisher consistent in a classification sense.

In the first step, let's prove that with probability 1 that $|f^*(x)| \le 1 + \varepsilon$. This step is identical to proving the Fisher Consistency of SVM. If a function $f(x)$ has a set $A_1$ with positive probability in $\mathcal{X}$ such that for $\forall x \in \mathcal{X}$, $|f(x)| > 1 + \varepsilon$, then we can truncate those values to $1 + \varepsilon$. In other word, consider $f^{new}(x) = f(x)$ for $x \in A_1{}^c$, $f^{new}(x) = (1 + \varepsilon)sgn(f(x))$ for $x \in A_1$. Then let's prove $f^{new}$ is better than $f$. We can see that the decision implied by $f$ and $f^{new}$ is the same. So the two constrains in problem (4) do not change. However, by looking at the objective function $E[(1 + \varepsilon - Yf(X))_+] = E[\eta(X)(1 + \varepsilon - f(X))_+ + (1 - \eta(X))(1 + \varepsilon + f(X))_+]$, we can see $f^{new}$ gives smaller loss for all the $X$ such that $\eta_X \ne 0, 1$, so that $f^*$ will give a smaller expected loss in $A_1$.

The next step, we prove $|f^*(x)| \ge \varepsilon$ in a similar way. If a function has a set $A_2$ with positive probability in $\mathcal{X}$ such that for $\forall x \in \mathcal{X}$, $|f(x)| < \varepsilon$, then we can enlarge those values of $|f|$ to $\varepsilon sgn(\eta(X) - \frac{1}{2})$. In other words, consider $f^{new}(x) = f(x)$ for $x \in A_2{}^c$, $f^{new}(x) = \varepsilon sgn(\eta(X) - \frac{1}{2})$ otherwise. Then let's prove $f^{new}$ is better than $f$. We can see that the decision implied by $f$ and $f^{new}$ is the same. So the two constrains in problem (4) do not change. However, by considering the objective function $E[(1 + \varepsilon - Yf(X))_+]$ and the result of first step we have $E[(1 + \varepsilon - Yf(X))_+] = E[(1 + \varepsilon - Yf(X))] = E[\eta(X)(1 + \varepsilon - f(X)) + (1 - \eta(X))(1 + \varepsilon + f(X))] = E[1 + \varepsilon + (1 - 2\eta(X))f(X)]$. Thus we have $E[H_\varepsilon(f)] - E[H_\varepsilon(f^{new})] = E[(1 - 2\eta(X))(f(X) - f^{new}(X))] = E[(1 - 2\eta(X))(f(X) - \varepsilon sgn(\eta(X) - \frac{1}{2}))1_{X \in A_2}] > 0$.

In the third step, we are going to show that $f^*$ is Fisher Consistent in the classic classification sense. In other words, $sgn(f^*(x)) = sgn(\eta(x) - \frac{1}{2})$ with probability 1. Because of symmetry, let's just prove the case that $\eta(X) > \frac{1}{2}$. If a function has a set $A_3$ with positive probability in $\mathcal{X}$ such that for $\forall x \in \mathcal{X}$, $|f(x)| < 0, \eta(x) > \frac{1}{2}$, then we can make them to $\varepsilon$. In other words, consider $f^{new}(x) = f(x)$ for $x \in A_3{}^c$, $f^{new}(x) = \varepsilon$ otherwise. Then let's prove $f^{new}$ is more efficient than $f$. The second constraint will not change since $\{x : f(x) > \varepsilon\} = \{x : f^{new}(x) > \varepsilon\}$. The second constraint is also satisfied by $f^{new}$ because we actually have $\{x : f^{new}(x) < -\varepsilon\} \subseteq \{x : f(x) < -\varepsilon\}$. However, $E[H_\varepsilon(f)] - E[H_\varepsilon(f^{new})] = E[(1 - 2\eta(X))(f(X) - \varepsilon)1_{X \in A_3}] > 0$.

In last step of part one, we want to prove that $f^*(x)$ does not take values between $\varepsilon$ and $\varepsilon + 1$ with probability 1. If a function has a set $A_4$ with positive probability in $\mathcal{X}$ such that for $\forall x \in \mathcal{X}$, $\varepsilon < |f(x)| < 1 + \varepsilon$, then we can enlarge those values of $f$ to $(1 + \varepsilon)sgn(\eta(X) - \frac{1}{2})$. In other words, consider $f^{new}(x) = f(x)$ for $x \in A_2{}^c$, $f^{new}(x) = (1 + \varepsilon)sgn(\eta(X) - \frac{1}{2})$ otherwise. Then let's prove $f^{new}$ is more efficient than $f$. By considering the result of step three, we

have the two constraints of $f$ is the same as $f^{new}$, because here we only need to consider the function $f$ such that $sgn(f(x)) = sgn(\eta(x) - \frac{1}{2})$. However, $E[H_\varepsilon(f)] - E[H_\varepsilon(f^{new})] = E[(1 - 2\eta(X))(f(X) - (1+\varepsilon)sgn(\eta(x) - \frac{1}{2}))1_{X \in A_4}] > 0$.

Now we have proved that $f^*$ only takes value of $1 + \varepsilon, \varepsilon, -\varepsilon, -(1+\varepsilon)$, with probability 1. That is to say $\mathcal{F}^*(1 + \varepsilon, \varepsilon, -\varepsilon, -(1+\varepsilon))$ is a complete class of the problem (4).

The difference between the proof of Lemma 1 and Theorem 1 is that $\mathcal{F}^*(1+\varepsilon, \varepsilon, -\varepsilon, -(1+\varepsilon))$ is not only the complete class of problem (4). Moreover, for any function $f$ such that there does not exist a $g$ with $f(x) - g(x) = 0$ with probability 1, by the argument above, there exist a $g_f$ that is better than $f$. We can than draw a conclusion that any solution of problem (4) is in $\mathcal{F}^*(1 + \varepsilon, \varepsilon, -\varepsilon, -(1+\varepsilon))$ (with a difference of probability 0).

The second part of the proof is to show part (ii) and (iii) of 2. This can be verified by direct calculation which is similar to the proof of Lemma 1. The last part of this proof is to find out the optimal $t$ and $t'$. The procedure is also analogous to proof of Lemma 1, thus is omitted here. As a result, 1 is the solution of problem (4) with probability 1. $\qquad \square$

## Proof of Theorem 2

We prove this Theorem in two steps. Firstly, we want to use excess risk of $\bar{R} = P(Yf(X) < -\varepsilon) + \frac{1}{2}P(|f(X)| \leq \varepsilon)$ to bound the excess ambiguity $R$. This can be formalized to a Lemma below.

**Lemma 3.** *Let $\hat{f}$ be another function that suffices the constraints in (3), then under Assumption 2, for any $\varepsilon \geq 0$, we have $\frac{1}{c}(\bar{R}(\hat{f}, \varepsilon) - \bar{R}(f^*, \varepsilon)) \geq R(\hat{f}, \varepsilon) - R(f^*, \varepsilon)$.*

To prove this, we need to further use another lemma which can be regarded as an extension of the theorem before.

**Lemma 4.** *There $\exists c > 0$ satisfies Assumption 2, then for any fixed $\varepsilon \geq 0$, $f^*$ is also a solution of the following optimization problem*

$$minimize \quad (\frac{1}{2} - c)P(|f(X)| \leq \varepsilon) + P(Yf(X) < -\varepsilon) \tag{3}$$
$$subject \ to \quad P_j(Yf(X) < -\varepsilon) \leq \alpha_j, \quad j = \pm 1.$$

The proof of this Lemma 4 is analogous to the proof of Lemma 1, thus is omitted here. Actually one can change the definition (ii) and (iii) in 2 by replacing $\frac{1}{2}$ with $\frac{1}{2} + c$ and also show that for $|\eta(x) - \frac{1}{2}| \leq c$, $|f(x)| \leq \varepsilon$.

By Lemma 4, we have

$$\frac{1}{c}(\bar{R}(\hat{f}, \varepsilon) - \bar{R}(f^*, \varepsilon)) - (R(\hat{f}, \varepsilon) - R(f^*, \varepsilon))$$
$$= \frac{1}{c}(P(Y\hat{f}(X) \leq \varepsilon) - P(Yf^*(X) \leq \varepsilon)) - (P(|Y\hat{f}(X)| \leq \varepsilon) - P(|Yf^*(X)| \leq \varepsilon))$$
$$= \frac{1}{c}(((\frac{1}{2} - c)P(|\hat{f}(X)| \leq \varepsilon) + P(Y\hat{f}(X) < -\varepsilon)) - (\frac{1}{2} - c)P(|f^*(X)| \leq \varepsilon) + P(Yf^*(X) < -\varepsilon))$$
$$\geq 0$$

$\square$

The next step is the prove we can use the excess $R_H$ risk to bound the excess risk of $\bar{R}$, which gives the Lemma below.

**Lemma 5.** *Under Assumption 2, for any $f$ satisfies the constraints in (3), we have*

$$C\left(R_H(f,\varepsilon) - R_H(f^*,\varepsilon)\right) \geq \left(\bar{R}\left(f,\varepsilon\right) - \bar{R}\left(f^*,\varepsilon\right)\right) \tag{4}$$

*where $C = \frac{1}{4c} + \frac{1}{2}$.*

The proof consists of two steps. First, we will show that we only need to consider the $f$ which takes those values: $1+\varepsilon$, $\varepsilon+$, $\varepsilon$, $-\varepsilon$, $-\varepsilon-$, $-(1+\varepsilon)$. Here $\varepsilon+$ can be regarded as $\varepsilon$ plus a arbitrarily small number and it is similar for $-\varepsilon-$. This can be shown by direct calculation.

Assume $f : \mathcal{X} \to \mathcal{R}$ is an arbitrary discriminate function. Then we consider another function $\bar{f}(\boldsymbol{x}) = (1+\varepsilon)\mathbb{1}\left[f(\boldsymbol{x}) > \varepsilon, \eta(\boldsymbol{x}) \geq \frac{1}{2}\right] + (\varepsilon+)\mathbb{1}\left[f(\boldsymbol{x}) > \varepsilon, \eta < \frac{1}{2}\right] + \varepsilon\mathbb{1}\left[|f(\boldsymbol{x})| \leq \varepsilon, \eta \geq \frac{1}{2}\right] + (-\varepsilon)\mathbb{1}\left[|f(\boldsymbol{x})| \leq \varepsilon, \eta < \frac{1}{2}\right] + (-\varepsilon-)\mathbb{1}\left[f(\boldsymbol{x}) < -\varepsilon, \eta \geq \frac{1}{2}\right] + (-(1+\varepsilon))\mathbb{1}\left[f(\boldsymbol{x}) < -\varepsilon, \eta(\boldsymbol{x}) < \frac{1}{2}\right]$. It is easy to see $\phi_{(f,\varepsilon)} = \phi_{(\bar{f},\varepsilon)}$ so that $\bar{R}(f,\varepsilon) - \bar{R}(f^*,\varepsilon) = \bar{R}(\bar{f},\varepsilon) - \bar{R}(f^*,\varepsilon)$. Moreover, by direct calculation, one can show that $R_H(f(\boldsymbol{x})) \geq R_H(\bar{f}(\boldsymbol{x}))$ for all $\boldsymbol{x}$. So we can see change $f$ to $\bar{f}$ will always leads to a smaller excess surrogate risk while keep the excess risk the same.

The second part is to explicitly calculate the left hand side and the right hand side and show that the $C$ in the theorem really works. To simplify the notation, we give divide $\mathcal{X}$ by value of $f$ (now $f$ take 6 values). For instance, we define $S_{\varepsilon+1} = \{\boldsymbol{x} : f(\boldsymbol{x}) = \varepsilon + 1\}$ and by the first part, we can assume $\eta(S_{\varepsilon+}) < \frac{1}{2}$ and $\eta(S_{-\varepsilon-}) > \frac{1}{2}$. To ease the notation, we omit the independent variable $X$ in following expressions although the expectation is really taken with respect to it. Then we have

$$
\begin{aligned}
R_H(f) - R_H(f^*) = {} & E(\mathbb{1}[S_{\varepsilon+1}](2(1+\varepsilon)(1-\eta))) + E(\mathbb{1}[S_\varepsilon](1+2\varepsilon(1-\eta))) \\
& + E(\mathbb{1}[S_{-\varepsilon-}](1+2\varepsilon\eta)) + E(\mathbb{1}[S_{\varepsilon+}](1+2\varepsilon(1-\eta))) \\
& + E(\mathbb{1}[S_{-\varepsilon}](1+2\varepsilon\eta)) + E(\mathbb{1}\left[S_{-(\varepsilon+1)}\right](2(1+\varepsilon)\eta)) \\
& - E(\mathbb{1}\left[\frac{1}{2} \leq \eta \leq t_{-1}\right](1+2\varepsilon(1-\eta))) - E(\mathbb{1}\left[t_1 \leq \eta \leq \frac{1}{2}\right](1+2\varepsilon\eta)) \\
& - E(\mathbb{1}[\eta > t_{-1}](2(1+\varepsilon)(1-\eta))) - E(\mathbb{1}[\eta < t_1](2(1+\varepsilon)\eta))
\end{aligned}
$$

and

$$
\begin{aligned}
R(f) - R(f^*) = {} & E(\mathbb{1}[S_{\varepsilon+1}](1-\eta)) + E(\mathbb{1}[S_\varepsilon](\frac{1}{2})) + E(\mathbb{1}[S_{-\varepsilon-}](\eta) \\
& + E(\mathbb{1}[S_{\varepsilon+}](1-\eta)) + E(\mathbb{1}[S_{-\varepsilon}](\frac{1}{2})) + E(\mathbb{1}\left[S_{-(\varepsilon+1)}\right](\eta)) \\
& - E(\mathbb{1}\left[\frac{1}{2} \leq \eta \leq t_{-1}\right](\frac{1}{2})) - E(\mathbb{1}\left[t_1 \leq \eta \leq \frac{1}{2}\right](\frac{1}{2})) \\
& - E(\mathbb{1}[\eta > t_{-1}](1-\eta)) - E(\mathbb{1}[\eta < t_1](\eta))
\end{aligned}
$$

Then by some algebra, we have $C(R_H(f) - R_H(f^*)) \geq R(f) - R(f^*)$ is equivalent to $A + 2\varepsilon CB \geq 0$ where

$$
\begin{aligned}
A =\ & E(\mathbb{1}[S_{\varepsilon+1}]((2C-1)(1-\eta))) + E(\mathbb{1}[S_\varepsilon](C-\frac{1}{2})) \\
& + E(\mathbb{1}[S_{-\varepsilon-}](C-\eta)) + E(\mathbb{1}[S_{\varepsilon+}](C-(1-\eta))) \\
& + E(\mathbb{1}[S_{-\varepsilon}](C-\frac{1}{2})) + E(\mathbb{1}\big[S_{-(\varepsilon+1)}\big]((2C-1)\eta)) \\
& - E(\mathbb{1}\Big[\frac{1}{2} \leq \eta \leq t_{-1}\Big](C-\frac{1}{2})) - E(\mathbb{1}\Big[t_1 \leq \eta \leq \frac{1}{2}\Big](C-\frac{1}{2})) \\
& - E(\mathbb{1}[\eta > t_{-1}]((2C-1)(1-\eta))) - E(\mathbb{1}[\eta < t_1]((2C-1)\eta))
\end{aligned}
$$

and $B = P(f(\boldsymbol{X})Y < 0) - P(f^*(\boldsymbol{X})Y < 0)$. By the definition of $f^*$ we can easily see that $B \geq 0$. So the rest is to show $A \geq 0$. We can only focus on $C > \frac{1}{2}$. Divide A by $2C-1$ and do some algebra, we have $A \geq 0$ is equivalent to

$$
\begin{aligned}
& (E(\mathbb{1}[S_{\varepsilon+1}]((1-\eta)) + E(\mathbb{1}[S_\varepsilon](\frac{1}{2}) + E(\mathbb{1}[S_{-\varepsilon-}](\frac{1}{2}) \\
& + E(\mathbb{1}[S_{\varepsilon+}](\frac{1}{2})) + E(\mathbb{1}[S_{-\varepsilon}](\frac{1}{2}) + E(\mathbb{1}\big[S_{-(\varepsilon+1)}\big](\eta)))) \\
& - (E(\mathbb{1}\Big[\frac{1}{2} \leq \eta \leq t_{-1}\Big](\frac{1}{2})) + E(\mathbb{1}\Big[t_1 \leq \eta \leq \frac{1}{2}\Big](\frac{1}{2}) \\
& + E(\mathbb{1}[\eta > t_{-1}](1-\eta) - E(\mathbb{1}[\eta < t_1](\eta)))) \\
& \geq \frac{1}{2C-1}(E(\mathbb{1}[S_{-\varepsilon-}](\eta - \frac{1}{2}) + E(\mathbb{1}[S_{\varepsilon+}](\frac{1}{2} - \eta))))
\end{aligned}
$$

It is not hard to see the first part of the left hand side is a $\bar{R}$ risk of a classifier with $+1$ prediction at $S_{\varepsilon+1}$, negative prediction at $S_{-(\varepsilon+1)}$ and ambiguity else where. The second part is the risk of $f^*$. By definition of $f^*$, we have $P_{-1}(\eta \leq t_{-1}) = 1 - \alpha_{-1}$, $P_1(\eta \geq t_1) = 1 - \alpha_1$. Let $\alpha'_{-1} = \alpha_{-1} - P_{-1}(S_{\varepsilon+})$ and $\alpha'_1 = \alpha_1 - P_1(S_{-\varepsilon-})$ and let $t'_{-1}$ and $t'_1$ satisfy $P_{-1}(\eta \leq t'_{-1}) = 1 - \alpha'_{-1}$, $P_1(\eta \geq t'_1) = 1 - \alpha'_1$. Because $\eta(S_{\varepsilon+}) < \frac{1}{2}$, we have $P(t_{-1} < \eta \leq t'_{-1}) > P(S_{\varepsilon+})$ by Bayes Formula. Similarly $P(t'_1 \leq \eta < t_1) > P(S_{-\varepsilon-})$. So at last, we have

$$
\begin{aligned}
\text{LHS of above} & \geq E(\mathbb{1}\big[t_{-1} < \eta \leq t'_-\big](\eta - \frac{1}{2})) + E(\mathbb{1}\big[t'_+ \leq \eta < t_1\big](\frac{1}{2} - \eta)) \\
& \geq c(P(t_{-1} < \eta \leq t'_-) + P(t'_+ \leq \eta < t_1)) \geq c(P(S_{\varepsilon+}) + P(S_{-\varepsilon-})) \\
& = \frac{1}{2C-1}\frac{1}{2}(P(S_{\varepsilon+}) + P(S_{-\varepsilon-})) \\
& \geq \frac{1}{2C-1}(E(\mathbb{1}[S_{-\varepsilon-}](\eta - \frac{1}{2}) + E(\mathbb{1}[S_{\varepsilon+}](\frac{1}{2} - \eta))))
\end{aligned}
$$

So we have $A \geq 0$ thus the statement of our theorem holds.

Note that one can induce a small $\delta$, i.e., using $\varepsilon + \delta$ instead of using the notation $\varepsilon+$ and let $\delta$ goes to 0 at the end of the proof to make it more rigorous. However, because there is no limit involved in other parts of this proof, we can live with this notation to keep us from

those trouble. □

Lastly, Theorem 2 is a direct corollary of Lemma 3 and Lemma 5. □

## Proof of Theorem 3

To prove this theorem, we need to introduce Rademacher complexity which has been widely used in statistical machine learning theory.

Here we only prove inequality for $Y = -1$, the proof for $Y = 1$ case can be down analogously. Without loss of generality, we assume the first $n_{-1}$ observations are from $-1$.

Let $\sigma = \{\sigma_i; i = 1, ..., n_{-1}\}$ be independent and identically distributed random variables from discrete uniform distribution U({-1,1}). Also denote by $S$ a sample of observations $(\boldsymbol{x_i}, y_i)$; i=1,...,$n_{-1}$, independent and identically distributed from the underlying distribution $P(\boldsymbol{X}, Y | Y = -1)$ (Y will always be $-1$ in this case). we define the empirical Rademacher complexity of the function class with fixed b, $\mathcal{H}_K^b(s) = \{h(x) + b | h \in \mathcal{H}_K, ||h||_{\mathcal{H}_K} \leq s\}$ as follows,

$$\hat{R}_{n_{-1}}\{\mathcal{H}_K^b(s)\} = E_\sigma[\sup_{f \in \mathcal{H}_K^b(s)} \frac{1}{n_{-1}} \sum_{i=1}^{n_{-1}} \sigma_i H_{-\varepsilon}(y_i f(\boldsymbol{x_i}))] \qquad (5)$$

Here $E_\sigma$ means taking expectation with respect to the joint distribution of $\sigma$. Moreover, we can define the Rademacher complexity of $\mathcal{H}_K^b(s)$ to be

$$R_{n_{-1}}\{\mathcal{H}_K^b(s)\} = E_{\sigma,S}[\sup_{f \in \mathcal{H}_K^b(s)} \frac{1}{n_{-1}} \sum_{i=1}^{n_{-1}} \sigma_i H_{-\varepsilon}(y_i f(\boldsymbol{X_i}))] \qquad (6)$$

where S is the sample space given $Y = -1$.

The next step is to construct the standard inequality of Rademacher complexity. It controls the expected hinge loss for negative group by the summation of empirical hinge loss, empirical Rademacher complexity and a small penalty term, which can be summarized in the following lemma. This lemma is important and will be used in the proves follows.

**Lemma 6.** *Let* $\hat{R}_n\{\mathcal{H}_K^b(s)\}$ *and* $R_n\{\mathcal{H}_K^b(s)\}$ *be defined as above. Then with probability at least* $1 - \zeta$,

$$E(H_{-\varepsilon}(Y f(\boldsymbol{X}))) \leq \frac{1}{n_{-1}} \sum_{i=1}^{n_{-1}} H_{-\varepsilon}(y_i f(\boldsymbol{x_i})) + 2R_{n_{-1}}\{\mathcal{H}_K^b(s)\} + T_{n_{-1}}(\zeta), \qquad (7)$$

*Moreover, with probability at least* $1 - \zeta$,

$$E(H_{-\varepsilon}(Y f(\boldsymbol{X}))) \leq \frac{1}{n_{-1}} \sum_{i=1}^{n_{-1}} H_{-\varepsilon}(y_i f(\boldsymbol{x_i})) + 2\hat{R}_{n_{-1}}\{\mathcal{H}_K^b(s)\} + 3T_{n_{-1}}(\zeta/2). \qquad (8)$$

*Proof.* The proof consist of three parts. In the first part, we use the McDiarmid inequality to bound the left hand side of inequality 7 by its empirical counterpart and $\phi(S)$ which is

define below:

$$\phi(S) = \sup_{f \in \mathcal{H}_K^b(s)} \{E(H_{-\varepsilon}(Yf(X))) - \frac{1}{n-1} \sum_{i=1}^{n-1} H_{-\varepsilon}(y_i f(\boldsymbol{x_i}))\}$$

Let $S^{(i)} = \{(\boldsymbol{x_1}, y_1), ...(\boldsymbol{x_i}', y_i), ...(\boldsymbol{x_n}, y_n)\}$ be another sample from $P(\boldsymbol{X}, Y | Y = -1)$, where the difference between $S$ and $S^{(i,x)}$ is just the $i$th observation. Then by definition, we have

$$\phi(S) - \phi(S^{(i)}) = \sup_{f \in \mathcal{H}_K^b(s)} \{E(H_{-\varepsilon}(Yf(X))) - \frac{1}{n-1} \sum_S H_{-\varepsilon}(y_i f(\boldsymbol{x_i}))\}$$
$$- \sup_{f \in \mathcal{H}_K^b(s)} \{E(H_{-\varepsilon}(Yf(X))) - \frac{1}{n-1} \sum_{S^{i,\boldsymbol{x}}} H_{-\varepsilon}(y_i f(\boldsymbol{x_i}))\}.$$

Note that it is easy to show the difference of supremum of two functions is smaller than the supremum of the difference of two functions.

Then we have

$$\phi(S) - \phi(S^{(i)}) \leq \sup_{f \in \mathcal{H}_K^b(s)} \{E(H_{-\varepsilon}(Yf(X))) - \frac{1}{n-1} \sum_S H_{-\varepsilon}(y_i f(\boldsymbol{x_i}))\}$$
$$- \{E(H_{-\varepsilon}(Yf(X))) - \frac{1}{n-1} \sum_{S^{i,\boldsymbol{x}}} H_{-\varepsilon}(y_i f(\boldsymbol{x_i}))\}$$
$$= \sup_{f \in \mathcal{H}_K^b(s)} \{\frac{1}{n-1} H_{-\varepsilon}(y_i f(\boldsymbol{x_i})) - H_{-\varepsilon}(y_i f(\boldsymbol{x_i'}))\}$$
$$\leq \sup_{f \in \mathcal{H}_K^b(s)} \{\frac{1}{n-1} |\{f(\boldsymbol{x_i}) - f(\boldsymbol{x_i'})|\}$$
$$\leq \sup_{h \in \mathcal{H}_K, ||h||_{\mathcal{H}_K} \leq s} \{\frac{1}{n-1} |\{h(\boldsymbol{x_i}) - h(\boldsymbol{x_i'})|\}$$
$$\leq \sup_{h \in \mathcal{H}_K, ||h||_{\mathcal{H}_K} \leq s} \{\frac{1}{n-1} |\langle h, K(\boldsymbol{x_i}, \cdot) \rangle - \langle h, K(\boldsymbol{x_i'}, \cdot) \rangle|\}$$
$$\leq \frac{1}{n-1} \sup_{h \in \mathcal{H}_K, ||h||_{\mathcal{H}_K} \leq s} \{|\langle h, K(\boldsymbol{x_i}, \cdot) \rangle| + |\langle h, K(\boldsymbol{x_i'}, \cdot) \rangle|\}$$
$$\leq \frac{2}{n-1} \sup_{h \in \mathcal{H}_K, ||h||_{\mathcal{H}_K} \leq s, \boldsymbol{x} \in \mathcal{X}} \{\sqrt{||h||_{\mathcal{H}_K} ||K(\boldsymbol{x}, \boldsymbol{x})||}\}$$
$$\leq \frac{2\sqrt{sr}}{n-1}$$

Because $S$ and $S^i$ are symmetric, as a result, we have $|\phi(S) - \phi(S^{(i)})| \leq \frac{2\sqrt{sr}}{n-1}$.

Next, by the McDiarmid inequality, we have that for any $t > 0$, $P(\phi - E(\phi(S)) \geq t) \leq exp(-\frac{t^2 n_{-1}}{2sr})$, or equivalently, with probability $1 - \zeta$, $\phi(S) - E(\phi(S)) \leq T_n(\zeta)$. Consequently, we have that with probability at least $1 - \zeta$, $E(H_{-\varepsilon}(Yf(\boldsymbol{X}))) \leq \frac{1}{n-1} \sum_{y_i=-1} H_{-\varepsilon}(y_i f(\boldsymbol{x_i})) +$

$E\{\phi(S)\} + T_{n_{-1}}(\zeta)$. This gives the first part of the proof.

In the second part, we need to bound $E\{\phi(S)\}$ by the Rademacher complexity. Define $S' = \{(x_i', y_i'); i = 1, ..., n_{-1}\}$ as an independent identical duplicate of $S$. Then we have that

$$E\{\phi(S)\} = E_S(\sup_{f \in \mathcal{H}_K^b(s)} E_{S'}[\frac{1}{n_{-1}} \sum_{S'} H_{-\varepsilon}(y_i' f(\boldsymbol{x_i'})) - \frac{1}{n_{-1}} \sum_{S} H_{-\varepsilon}(y_i f(\boldsymbol{x_i}))]|S)$$

$$\leq E_{S,S'}[\frac{1}{n_{-1}} \sum_{S'} H_{-\varepsilon}(y_i' f(\boldsymbol{x_i'})) - \frac{1}{n_{-1}} \sum_{S} H_{\varepsilon}(y_i f(\boldsymbol{x_i}))]$$

$$= E_{S,S',\sigma}[\frac{1}{n_{-1}} \sum_{S'} \sigma_i H_{-\varepsilon}(y_i' f(\boldsymbol{x_i'})) - \frac{1}{n_{-1}} \sum_{S} \sigma_i H_{\varepsilon}(y_i f(\boldsymbol{x_i}))]$$

$$\leq 2R_{n_{-1}}\{\mathcal{H}_K^b(s)\}$$

Combining the first and second step, we have already proved first inequality in Lemma 6.

The third step is analogous to the first step. We will use the empirical Rademacher complexity to bound the population Rademacher complexity.

This can be shown by defining $\psi(S) = \hat{R_{n_{-1}}}\{\mathcal{H}_K^b(s)\}$ and it is easy to see $|\psi(S) - \psi(S')| \leq \frac{2\sqrt{sr}}{n_{-1}}$ by the definition of empirical Rademacher complexity. Then we can use McDiarmid inequality again and get with probability at least $1 - \zeta$, $\psi(S) - E(\psi(S)) \leq T_{n_{-1}}(\zeta)$. At last, we can combine this outcome and 7 by choose the confidence level to be $1 - \zeta/2$ to get 8. $\square$

Then last step will be controlling the empirical Rademacher complexity for kernel learning. In particular, by Lemma 4.2 and Theorem 5.5 in Mohri et al. (2012), we can have that $\hat{R_{n_{-1}}}\{\mathcal{H}_K^b(s)\}$ can be upper bounded by the following inequality

$$\hat{R_{n_{-1}}}\{\mathcal{H}_K^b(s)\} \leq E_\sigma[\sup_{h \in \mathcal{H}_K, ||h||_{\mathcal{H}_K} \leq s} \frac{1}{n_{-1}} \sum_{y_i = -1} \sigma_i h(\boldsymbol{x_i})]$$

$$\leq \frac{rs}{\sqrt{n_{-1}}}.$$

$\square$

## Proof of Theorem 4

This proof is similar to proof of Theorem 5 in Rigollet and Tong (2011).

Statement (a) of this theorem is the direct Corollary of Theorem 3. One can see the proof for Lemma 6 does not only work for $H_{-\varepsilon}$, but also work for $H_c$ with any $c$. In particular, it works for $\varepsilon$. So define the events $E_1$ and $E_2$. Let $R_H^j(f, c) = E(H_c(Yf(X))|Y = j)$, and $\hat{R}_H^j(f, c)$ be its empirical counterpart for $j = \pm 1$.

$$E^-(f, \varepsilon) = \{|\hat{R}_H^{-1}(f, -\varepsilon) - R_H^{-1}(f, -\varepsilon)| \leq \frac{\kappa}{\sqrt{n_{-1}}}, |\hat{R}_H^1(f, -\varepsilon) - R_H^1(f, -\varepsilon)| \leq \frac{\kappa}{\sqrt{n_1}}\},$$

$$E^+(f,\varepsilon) = \{|\hat{R}_H^{-1}(f,\varepsilon) - R_H^{-1}(f,\varepsilon)| \le \frac{\kappa}{\sqrt{n_{-1}}}, |\hat{R}_H^1(f,\varepsilon) - R_H^1(f,\varepsilon)| \le \frac{\kappa}{\sqrt{n_1}}\}.$$

By Theorem 3, $P(E^-) \ge 1 - 2\zeta$ and $P(E^+) \ge 1 - 2\zeta$ for any fix $f \in \mathcal{H}_K(s)$ and $c \in \mathbb{R}$.

To study statement (b), we can decompose the left hand side of the inequality into three parts and study them one by one.

$$R_H(\hat{f}, \hat{\varepsilon}) - \min_{(f,\varepsilon) \in \mathcal{F}(0,s)} R_H(f,\varepsilon) = A_1 + A_2 + A_3$$

Where

$$A_1 = (R_H(\hat{f}, \hat{\varepsilon}) - \hat{R}_H(\hat{f}, \hat{\varepsilon})) + (\hat{R}_H(\hat{f}, \hat{\varepsilon}) - \min_{(f,\varepsilon) \in \hat{\mathcal{F}}(\kappa,s)} R_H(f,\varepsilon))$$

$$A_2 = \min_{(f,\varepsilon) \in \hat{\mathcal{F}}(\kappa,s)} R_H(f,\varepsilon) - \min_{(f,\varepsilon) \in \mathcal{F}(2\kappa,s)} R_H(f,\varepsilon)$$

$$A_3 = \min_{(f,\varepsilon) \in \mathcal{F}(2\kappa,s)} R_H(f,\varepsilon) - \min_{(f,\varepsilon) \in \mathcal{F}(0,s)} R_H(f,\varepsilon)$$

Because we only focus on $E_1 = E^+(\hat{f}, \hat{\varepsilon}) \bigcap E^+(\mathrm{argmin}_{(f,\varepsilon) \in \hat{\mathcal{F}}(\kappa,s)} R_H(f,\varepsilon))$, then we have

$$A_1 \le \frac{2\kappa}{\sqrt{n}}$$

It is easy to see that $A_2 \le 0$ for large enough n on $E_2 = E^-(\mathrm{argmin}_{(f,\varepsilon) \in \mathcal{F}(2\kappa,s)} R_H(f,\varepsilon))$.

The last part of the proof is to bound $A_3$. To begin with the proof, let's first introduce a lemma.

**Lemma 7.** *Let $\gamma_s((\alpha_{-1}, \alpha_1)) = $ be a function from $[0,1]^2$ to $\inf_{f \in \mathcal{F}(0,s)} R_H(f,\varepsilon)$. Then $\gamma_\varepsilon$ is convex in $[0,1]^2$. Moreover, $\gamma_s((\alpha_{-1}, \alpha_1)) \le \gamma_s((\alpha'_{-1}, \alpha'_1))$ for $\alpha_{-1} \ge \alpha'_{-1}$ and $\alpha_1 \ge \alpha'_1$.*

*Proof.* By the convexity of loss function $H_c$, we have $E(H_{(\theta c_1 + (1-\theta)c_2)}(Y(\theta f_1(X) + (1-\theta)f_2(X)))) \le \theta E(H_{c_1}(Y f_1(X))) + (1-\theta)E(H_{c_2}(Y f_2(X)))$ for all $\theta \in [0,1]$. By definition of infimum, for any $\mu > 0$ and $\alpha^1 = (\alpha_{-1}, \alpha_1)$, there exists a $f_1 \in \mathcal{F}(0,s)$, such that $\gamma_s(\alpha^1) > E(H_\varepsilon(Y f_1(X))) - \mu$, and for another $\alpha^2$, there exists a $f_2$ as well.

By the argument above, we have $\gamma_s(\theta \alpha^1 + (1-\theta)\alpha^2) \le E(H_{(\theta \varepsilon_1 + (1-\theta)\varepsilon_2)}(Y(\theta f_1(X) + (1-\theta)f_2(X)))) \le \theta E(H_{\varepsilon_1}(Y f_1(X))) + (1-\theta)E(H_{\varepsilon_2}(Y f_2(X))) \le \theta \gamma_s(\alpha^1) + (1-\theta)\gamma_s(\alpha^2) - \mu$ for all positive $\mu$. And it is easy to verify that $\theta f_1 + (1-\theta)f_2$ and $(\theta \varepsilon_1 + (1-\theta)\varepsilon_2)$ give satisfy the constraints. So that $\gamma_s$ is convex.

The second statement of the lemma is easy to see by noticing that $\mathcal{H}_K(s) \cap \mathcal{F}_\varepsilon(\alpha'_{-1}, \alpha'_1) \subset \mathcal{H}_K(s) \cap \mathcal{F}_\varepsilon(\alpha_{-1}, \alpha_1)$ for $\alpha_{-1} \ge \alpha'_{-1}$ and $\alpha_1 \ge \alpha'_1$. □

The last part of the proof is from the convexity of $\gamma_s$. For a large enough $n_{-1}$ and $n_1$, we will finally have $\frac{\kappa}{\sqrt{n_{-1}}} < \alpha_{-1}, \frac{\kappa}{\sqrt{n_1}} < \alpha_1)$. Let $\nu = (\frac{\kappa}{\sqrt{n_{-1}}}, \frac{\kappa}{\sqrt{n_1}})$.

Now by convexity of $\gamma_s$, we have $\gamma_s(\alpha) - \gamma_s(\alpha - \nu) \ge \nu \cdot g$ and $\gamma_s(\alpha - \nu_0) - \gamma_s(\alpha - \nu) \ge (\nu - \nu_0) \cdot g$, where g in any member of the subgradiant of $\gamma$ at $\alpha - \nu$. After combining these two inequalities, we have

$$\gamma_s((0,0)) - \gamma_s(\alpha - \nu) \ge (\alpha - \nu) \cdot (-g)$$

$$\geq \min \Big\{ \frac{\alpha_{-1} - \frac{\kappa}{\sqrt{n_{-1}}}}{\frac{\kappa}{\sqrt{n_{-1}}}}, \frac{\alpha_1 - \frac{\kappa}{\sqrt{n_1}}}{\frac{\kappa}{\sqrt{n_1}}} \Big\} \nu \cdot (-g)$$

$$\geq \min \Big\{ \frac{\alpha_{-1} - \frac{\kappa}{\sqrt{n_{-1}}}}{\frac{\kappa}{\sqrt{n_{-1}}}}, \frac{\alpha_1 - \frac{\kappa}{\sqrt{n_1}}}{\frac{\kappa}{\sqrt{n_1}}} \Big\} (\gamma_s(\alpha - \nu) - \gamma_s(\alpha))$$

This will finally lead us to $\gamma_s(\alpha - \nu) - \gamma_s(\alpha) \leq (\gamma_s(\alpha) - \gamma_s(\alpha - \nu)) \frac{2 \max\{\frac{\kappa}{\sqrt{n_{-1}}}, \frac{\kappa}{\sqrt{n_1}}\}}{\min\{\alpha_{-1}, \alpha_1\}} \leq \frac{2\kappa}{\min\{\alpha_{-1}, \alpha_1\} \min\{\sqrt{n_{-1}}, \sqrt{n_1}\}}$. The last inequality is directly from the fact that $f(x) \equiv 0$ and $\varepsilon = 1$ satisfies the constraints of problem (5) and gives ambiguity loss 2. Thus we have $A_3 \leq \frac{4\kappa}{\min\{\alpha_{-1}, \alpha_1\} \min\{\sqrt{n_{-1}}, \sqrt{n_1}\}}$.

Then the proof is finished by combining $A_1, A_2, A_3$. $\hfill\square$

## 2 More on numerical study

For each simulation scenario, we give a plot of non-coverage rates for both $-1$ and 1 class. We also give plots of the proportion of instances in which both classes have the desired test non-coverage rates, e.g. 0.05 or smaller.

### Linear model with nonlinear Bayes rule

Figure 1: Non-coverage rates for all the models. We can see that weighted CSVM has a smaller non-coverage rates when sample size become larger, which explains why it has a relatively larger ambiguity. It worth to note that when $n = 80$, weighted CSVM has a significantly smaller non-coverage rates than plug-in methods and maintain a smaller (or comparable) ambiguity.

Figure 2: Success (to cover desired observations) rates for all the models. We can see that weighted CSVM has a greater success rates when sample size become larger, which also explains why it has a relatively larger ambiguity. It worth to note that when $n = 80$, weighted CSVM has a much larger non-coverage rates than plug-in methods and maintain a smaller (or comparable) ambiguity.

## Moderate dimensional polynomial boundary

Figure 3: Non-coverage rates for all the models. We can see that weighted CSVM and kNN has a smaller non-coverage rates when the other three have similar non-coverage rates. But within those two groups, the proposed model always has a smaller ambiguity.

Figure 4: Success rates for all the models. We can see that weighted CSVM and kNN has a larger success rates when the other three have similar success rates. But within those two groups, the proposed model always has a smaller ambiguity.

## High-dimensional donut

Figure 5: Non-coverage rates for all the models. We can see that weighted CSVM and kNN has a smaller non-coverage rates when the other three have similar non-coverage rates. But within those two groups, the proposed model always has a smaller ambiguity.

Figure 6: Success rates for all the models. We can see that weighted CSVM and kNN has a larger success rates when the other three have similar success rates. But within those two groups, the proposed model always has a smaller ambiguity..