[Reviews · NeurIPS 2018]

Reviewer 1



Interesting paper on an interesting topic. Contains theoretical results, a numerical algorithm, and numerical experiments. Perhaps some additional references on surrogate loss functions and their risk should be mentioned on page 4, line 128.

Reviewer 2



Summary The paper proposes an SVM-like classification method for estimating sets containing a pre-specified amount of samples for each class. The overlap of these two sets is a region with ambiguity and should thus be small. The key results are: problem formulation and reformulation using a convex surrogate loss function. Fisher consistency of this loss, and an inequality comparing excess risks. Finite sample bounds, a brief description of an algorithm, and experiments on both artificial and real-world data. Impression The problem formulation is very interesting and the combination of theoretical and experimental results is above standard. In addition, the paper is easy to follow. My main concerns are: - What is the conceptional difference between the proposed approach and classification with reject option as in [2]. It seems that both are equivalent by differently parametrized. Also, where is the difference to [13]? - The theoretical analysis in Section 4.2 is certainly not based on state-of-the-art techniques for analyzing SVMs. - The description of the algorithm is too short and its computational requirements are not discussed. - The experimental section should be improved: the type of SVM they use as a comparison is not explained at all. I guess, it is one with the hinge loss? If so, this cannot reliably estimate eta, or at least a monotone transform of parts of it. As a result a hinge loss SVM is simply not suitable for the considered problem and the brief hint to [13] in the middle of page 6 is not giving me any more information how to address this issue. As an alternative one could have either used a least-squares-like SVM or one using a convex loss with reject option as in [2]. Similarly, it is not clear, whether and how the other methods considered in the experiments have been adapted to the new problem formulation. here are some thoughts to the authors' reply: 1. there is a difference between the motivation of an approach the the resulting approach. I do agree that [2] has indeed a different motivation, but I actually would have loved to see a discussion on the resulting approach. For example, what effect have the subtle differences in the Bayes decision function in [2] and the present paper? I admit, however, that my concern/question was not too precise in this direction. 2. Again, I was a bit imprecise, but here is what I meant: analyzing (5) instead of (3) is a cheap trick, which comes with a price: either the translation requires an overly conservative s (if it works at all), or a data-dependent s. Modern methods (for oracle inequalities instead of generalization error bounds, though) can actually work implicitly with a data-dependent s, although they usually analyze (3) directly. But again, I should have been more precise. 3. Of course deriving the Langrangian is boring, but here is an aspect thats need to be discussed in more detail: line 171: The above optimization may be solved by any efficient quadratic programming routine. What are these EFFICIENT routines? Certainly not the ones used by the authors as their timing indicate. This hand-waiving argument really goes back to the nineties. 4 a) naive SVM is first mentioned on page 7 without any explanation there or later. This should be updated on page 6 b) "overall message" is a bit too much considering the few experiments. c) see 1, though I do agree that adjusting d is a problem Overall, I happy to raise my score by one based on the feedback

Reviewer 3



This paper provides an SVM-based set-valued classifier under the framework of classification with confidence''. This framework indeed has a long history in the literature of statistical learning and has recently regained attention due to practical concerns of classification problems such as class imbalance, asymmetric cost of wrong classification, etc. Strength: This paper puts SVM under the framework of classification with confidence. The idea of providing classification confidence set is important and practically useful, but the current methods rely on conditional probability estimation, which is itself a hard problem and may not be necessary for classification. Instead, SVM is one of the most practically successful classification algorithms. Therefore, making SVM available in this new framework will greatly increase the usefulness of both SVM and the framework of classification with confidence. Moreover, this paper clearly explains and motivated the formulation of the new optimization algorithm and provides convincing theoretical justification of the methodology. Questions and Comments: The clarity of the theoretical section can be improved, for example, by providing some more discussions and examples of the assumptions. In particular, in Assumption 3, do $\nu_1$ and $\nu_{-1}$ need to be constants, can they vanish to 0? Also, the analysis focuses on the constrained version but the algorithm is the penalized version. It is known in other cases (such as penalized linear regression) that constrained versions are easier to analyze. Can the author(s) comment on the theoretical property of the penalized version? Finally, the simulation uses unrealistically large sample sizes. Smaller (for example, a few hundred) sample sizes may be more relevant. Writing: There are some missing punctuations, especially after math displays. For example: before line 11, line 81, line 145, line 148, line 160, etc.